# Increased processing of SINE B2 ncRNAs unveils a novel type of transcriptome deregulation in amyloid beta neuropathology

Yubo Cheng[1,2,3,4†], Luke Saville[1,2,3,4†], Babita Gollen[1,2,3,4†], Christopher Isaac[1,2,3,4†], Abel Belay[1,2,3,4], Jogender Mehla[3], Kush Patel[1,2,3], Nehal Thakor[1,2,3], Majid H Mohajerani[3], Athanasios Zovoilis[1,2,3,4*]

[1]Department of Chemistry and Biochemistry, University of Lethbridge, Lethbridge, Canada; [2]Southern Alberta Genome Sciences Centre, University of Lethbridge, Lethbridge, Canada; [3]Canadian Centre for Behavioral Neuroscience, University of Lethbridge, Lethbridge, Canada; [4]Alberta RNA Research and Training Institute, University of Lethbridge, Lethbridge, Canada

**Abstract** The functional importance of many non-coding RNAs (ncRNAs) generated by repetitive elements and their connection with pathologic processes remains elusive. B2 RNAs, a class of ncRNAs of the B2 family of SINE repeats, mediate through their processing the transcriptional activation of various genes in response to stress. Here, we show that this response is dysfunctional during amyloid beta toxicity and pathology in the mouse hippocampus due to increased levels of B2 RNA processing, leading to constitutively elevated B2 RNA target gene expression and high *Trp53* levels. Evidence indicates that Hsf1, a master regulator of stress response, mediates B2 RNA processing in hippocampal cells and is activated during amyloid toxicity, accelerating the processing of SINE RNAs and gene hyper-activation. Our study reveals that in mouse, SINE RNAs constitute a novel pathway deregulated in amyloid beta pathology, with potential implications for similar cases in the human brain, such as Alzheimer's disease (AD).

*For correspondence:
athanasios.zovoilis@uleth.ca

†These authors contributed equally to this work

Competing interests: The authors declare that no competing interests exist.

## Introduction

The number of patients with Alzheimer's disease (AD) is expected to skyrocket in the upcoming years (*Cornutiu, 2015*). The exact molecular mechanisms underlying AD are not fully understood, a fact that is underlined by the inauspicious results of recent clinical trials for potential therapeutic agents (*Hane et al., 2017*; *Cummings et al., 2019*). Amyloid pathology, and particularly, amyloid beta peptides and their aggregated forms have been connected with AD pathogenesis (*Bloom, 2014*) as well as with neurotoxicity in mouse models of amyloid pathology (*Ittner et al., 2010*). Nevertheless, the transcriptome changes involved in cell stress response to amyloid toxicity in brains with extensive amyloid beta pathology are still not entirely clear. The hippocampus is a primary target of amyloid pathology in humans. In healthy hippocampi, among the genes that have been implicated in transcriptome-environment interactions are stress response genes (SRGs) (*Gallo et al., 2018*). These genes have been initially described in other biological contexts, such as thermal and oxidative stress, as pro-survival genes activated early after the application of a stress stimulus, such as heat shock, that help the cell overcome the stress condition (*Mahat et al., 2016*). However, in addition to the cellular response to stress, many SRGs were shown to have a central role in the function of the mouse hippocampus, by mediating cell signaling and genome-environment interactions. In particular, we and others have shown that in the healthy hippocampus during

neural response to environmental stimuli, SRGs, such as those of the MAPK pathway, are transiently activated during various hippocampal processes including learning and response to cellular stress (*Peleg et al., 2010*; *Yutsudo et al., 2013*; *Sananbenesi et al., 2003*). Activation of SRGs is followed by a transient upregulation of the pro-apoptotic factor *Trp53* (*Peleg et al., 2010*) and, subsequently, of a pro-apoptotic miRNA, *Mir34c*, which are transiently induced by and as a response to the activation of pro-survival SRGs (*Zovoilis et al., 2011*; *Yamakuchi and Lowenstein, 2009*). *Trp53* activates the expression of genes engaged in promoting cell death in response to multiple forms of cellular stress including *Mir34c* (*Yamakuchi and Lowenstein, 2009*). This miRNA acts transiently as a guard and fine tuner of the expression of many SRGs by targeting them, thus, creating a negative feedback regulatory loop that keeps SRG expression in healthy cells under strict control. This facilitates the return to the pro-stimulation state in approximately 3 hr after the application of the stimulus (*Zovoilis et al., 2011*). In contrast, hippocampi of mouse models of amyloid pathology and postmortem brains of human patients of AD are characterized by abnormally high *Mir34c* levels that subsequently can lead to prolonged high *Trp53* levels and neural death (*Zovoilis et al., 2011*; *Yamakuchi and Lowenstein, 2009*). Given that many SRGs are upstream regulators of *Trp53-Mir34c* activation (*Gao et al., 2010*), high *Trp53* and *Mir34c* levels in amyloid pathology implied a possible transcriptome deregulation of the pathways that involve SRGs but whether such a deregulation exists, and which is the mechanism underlying this, it remained unknown. Interestingly, in a recent publication, we showed that expression of a number of SRGs is regulated by a class of non-protein coding (non-coding) RNAs called B2 SINE RNAs (*Zovoilis et al., 2016*), raising the possibility that these non-coding RNAs may be a missing link in the pathways connecting amyloid beta toxicity with transcriptome changes in mouse hippocampus during amyloid pathology.

SINE non-coding RNAs are transcribed by repetitive small interspersed nuclear elements (SINEs), with the subclass B2-repetitive elements being one of the most frequent in mouse (*Kramerov and Vassetzky, 2011*). SINE B2 elements, which are retrotransposons present in hundreds of thousands of copies, are part of the non-protein coding genome, and over the long haul, they have been regarded as genomic parasites and 'junk DNA' with no function (*Karijolich et al., 2017*). However, SINE B2 elements can be transcribed by RNA Polymerase III into SINE B2 RNAs (*Kramerov and Vassetzky, 2011*) and a number of recent studies have revealed a key role for SINE B2 RNAs in cellular response to stress. In particular, studies from the J. Kugel and J. Goodrich labs have shown that during response to cellular stress, levels of SINE B2 RNAs increase and suppress the transcription of a number of housekeeping genes through binding of RNA Polymerase II (RNA Pol II), potentially facilitating the redirection of cell resources to pro-survival pathways (*Yakovchuk et al., 2009*). In addition, we have recently shown that SINE B2 RNAs mediate cellular response to stress through the regulation of pro-survival stress response genes by acting as transcriptional switches. In particular, in the pro-cellular stress state SINE B2 RNAs bind RNA Pol II at several SRGs and suppress their transcription. In this way, stalled or delayed RNA Pol II remains poised for a fast activation and ramp-up of transcription when needed. Upon application of a stress stimulus, SINE B2 RNAs, which have a self-cleavage activity that is accelerated by their interaction with a protein (Ezh2) (*Hernandez et al., 2020*), are processed into unstable fragments that lack the ability to bind and suppress RNA Pol II. This event releases the delayed or stalled RNA Pol II and enables fast transcriptional activation of stress response genes (*Zovoilis et al., 2016*).

The above findings have revealed a novel role in cellular function for processing of SINE B2 RNAs (hereafter referred simply as B2 RNAs) through the activation of a number of SRGs regulated by them (hereafter called B2-SRGs). However, an association of the processing and destabilization of B2 RNAs with any pathological cellular process remains unknown. Here, given the importance of SRGs in hippocampal neuronal function, we examine whether this newly described B2 RNA-SRG regulatory mechanism is linked with pathological processes, focusing on transcriptome response to amyloid beta toxicity and pathology. To this end, we investigate whether B2-SRGs are indeed deregulated during amyloid pathology, which would imply a role for B2 RNAs in this condition and we examine whether amyloid toxicity is connected with changes in B2 RNA processing. Subsequently, we investigate the further upstream molecular mechanisms underlying any potential deregulation of B2 RNA processing in response to amyloid toxicity in hippocampal neural cells.

## Results

### B2 RNA regulated SRGs (B2-SRGs) are enriched in neural functions

In a previous study, we have identified genomic locations that are subject to regulation by SINE B2 non-coding RNAs during response to thermal stress (heat shock) through binding and suppression of transcription by the RNA Pol II at the pre-stimulus state. Upon induction of cellular stress through the application of a stimulus, SRGs in these locations become activated through B2 processing and release of RNA Pol II suppression (*Figure 1A*; *Zovoilis et al., 2016*). A list of B2-SRGs at these locations is available in *Supplementary file 1*. Response to thermal stress (heat shock) has been used for years as a basic study model of cellular response to stress. Proteins and gene pathways initially identified in heat shock have been subsequently shown to play identical pro-survival roles in other biological systems. Thus, we questioned whether there are other known cellular functions, beyond response to heat shock, that are connected with B2-SRGs.

As shown in *Figure 1B*, after performing a tissue enrichment analysis to identify tissue terms that are over-represented in the list of our SRGs, we found a significant enrichment of neural tissue terms compared to other tissues in our list. Similarly, during Gene Ontology term enrichment analysis, cellular compartments closely related to neural functions top the list of enriched terms in these genes, including among the first 10 entries terms such as synapse, postsynaptic density and membrane, dendrites, neural projections, and pre-synaptic membrane (*Figure 1C*, *Supplementary file 1*). Most importantly, after performing the same analysis for Biological Processes GO terms, B2-SRGs were found to be enriched considerably in neural-function-related terms. GO terms such as learning, nervous system development, synaptic transmission, synapse receptor localization, and neurotransmitter transport were among the first 10 entries with the highest adjusted p value scores enriched in B2-SRGs (*Figure 1D*, *Supplementary file 1*).

Among the biological processes potentially affected by B2 RNAs that were identified above was learning. We and others have already shown that learning processes in the mouse hippocampus are connected with the transient activation of a number of learning-associated genes and pathways, including many known SRGs (*Peleg et al., 2010*). For this reason, we examined whether any learning-associated genes are among the binding targets of B2 RNAs. To this end, we compared the distribution of B2 RNA binding sites in the genome between learning-associated genes and all genes. Indeed, as shown in *Figure 1E*, B2 RNA-binding sites were found to be enriched in learning-associated genes compared to other genes. In particular, based on learning-associated genes previously identified in mouse hippocampus (*Peleg et al., 2010*), among the B2-SRGs (1684 genes), 102 genes (*Figure 1E*, *Supplementary file 1*) are associated with learning. In addition, biological process terms enriched in B2-SRGs included, among others, pathways implicated with response to cellular stress in neural cells, and various genes implicated with synaptic function (*Figure 1E*, *Supplementary file 1*).

This data suggests that B2-SRGs could potentially affect a wide spectrum of functions beyond heat shock including response to cellular stress in neural cells, synaptic function, and learning. All these biological processes are heavily impaired in AD. Given the role of hippocampus as a primary target of amyloid pathology in AD, we decided to expand our investigation towards the potential role of B2-SRGs regulation in this pathological process. We have previously shown that in hippocampus the pro-apoptotic miRNA *Mir34c*, which is induced among others by *Trp53* and controls it in a negative feedback loop manner (*Yamakuchi and Lowenstein, 2009*), targets many learning-associated genes in order to help to restore their expression and *Trp53* expression levels after application of a stimulus (*Figure 1F*; *Zovoilis et al., 2011*; *Yamakuchi and Lowenstein, 2009*). In both mouse models of amyloid beta pathology and AD patient brains, we found persistently high levels of *Mir34c*. High *Mir34c* levels in hippocampi with amyloid pathology are indicative of a transcriptome-wide deregulation of *Trp53* and associated genes, as many of these genes are either direct or indirect upstream regulators of *Mir34c* (*Figure 1F*; *Rokavec et al., 2014*). Since, as shown above, B2-SRGs include many learning genes as well as other genes involved in neural function, we questioned whether such transcriptome changes in response to amyloid pathology involve B2-SRGs.

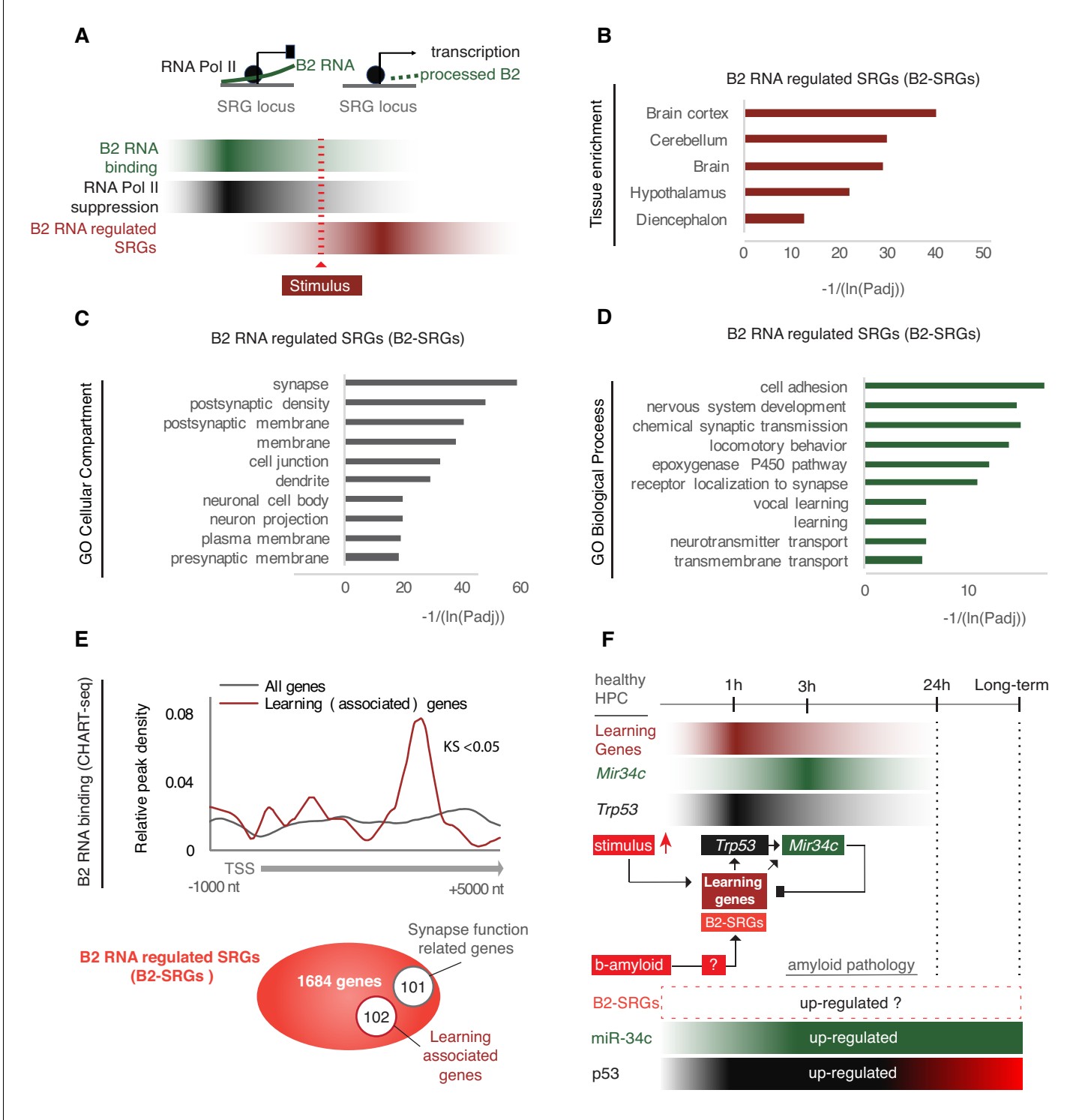

**Figure 1.** B2-SRGs are enriched in neural functions. (**A**) Regulation mode of SRGs by B2 RNA processing based on previous works (*Zovoilis et al., 2016*; *Yakovchuk et al., 2009*; *Ponicsan et al., 2010*). Color intensity represents higher B2 RNA binding (green), Pol II suppression (black) or SRG expression (red). (**B**) Tissue enrichment analysis of B2-SRGs listed in *Supplementary file 1* based on the DAVID functional annotation platform (*Huang et al., 2009a*; *Huang et al., 2009b*). The adjusted p-values of top five ranking terms are plotted as a function with higher scores representing lower adjusted p-values (higher statistical significance) ranging from padj 1.92E-04 for 'Diencephalon' to padj 8.91E-13 for 'Brain Cortex'. A complete list of all terms that pass the EASE 0.05 score threshold is presented in *Supplementary file 1*. (**C**) Gene ontology (GO) analysis (DAVID) of B2 RNA regulated genes on the basis of cellular compartments. The adjusted p-values of top 10 ranking terms are plotted as a function with higher scores representing lower adjusted p-values ranging from 2.23E-06 for 'Presynaptic membrane' to 1.88E-18 for 'Synapse'. A complete list of all terms that pass

*Figure 1 continued on next page*

*Figure 1 continued*

the EASE 0.05 score threshold is presented in *Supplementary file 1*. (D) GO analysis (DAVID) of B2 RNA regulated genes on the basis of biological processes. As above, adjusted p-values of top ranking terms are plotted as a function, with higher scores representing lower adjusted p-values ranging from 0.02 for 'Transmembrane transport' to 5.44E-06 for 'Cell adhesion'. A complete list of all terms that pass the EASE 0.05 score threshold is presented in *Supplementary file 1*. (E) Upper panel: Metagene analysis of distribution of genomic B2 RNA-binding sites across the start of genes from *Zovoilis et al., 2016*, comparing learning-associated genes from *Peleg et al., 2010* with all genes (Kolmogorov Smirnov test, KS <0.05). TSS represents the Transcription Start Site of these genes. Lower panel: Representation of the number of B2-SRGs that are (i) learning associated based on *Peleg et al., 2010* (*Supplementary file 1*) and (ii) synapse function related based on GO term enrichment (*Supplementary file 1*, right panel). (F) Regulatory loop of learning genes-*Trp53*-*Mir34c* in mouse hippocampus based on *Peleg et al., 2010*; *Zovoilis et al., 2011*; *Yamakuchi and Lowenstein, 2009*. A potential role of B2-SRGs, that include many learning genes, in affecting this loop in amyloid pathology remained unknown after these studies (noted with a question mark in the figure).

## A number of B2-SRGs get hyper-activated during the neurodegeneration phase of amyloid pathology

To test whether B2-SRGs are indeed deregulated in amyloid pathology, we employed a transgenic mouse model of amyloid pathology, APP$^{NL-G-F}$ (*Saito et al., 2014*), and the respective wild-type control (C57BL/6J). The same animal cohorts that were previously characterized through a battery of immunohistochemistry (IHC) and behavioral tests (*Mehla et al., 2019*) were used to isolate whole hippocampi for the transcriptome analysis conducted in the current study. *Figure 2A* depicts our experimental design while *Figure 2B* depicts the amyloid plaque deposition in the brains. The behavioral tests in these mouse cohorts are presented in our previous study (*Mehla et al., 2019*).

We have focused on three different mouse ages that correspond to different phases of amyloid beta pathology and represent the: (i) pre-symptomatic stage with undetectable (very low) amyloid plaque load (3 months - 3 m, *Figure 2B* left panels), (ii) stage of symptom manifestation (6 months - 6 m) that coincides with the active neurodegeneration phase and appearance of amyloid plaques (*Figure 2B*, middle-panels), and (iii) terminal stage of the pathology (12 months - 12 m, *Figure 2B* right panels) when mice have already acquired the extensive brain atrophy due to neural cell death. Whole hippocampi from mice of these three groups were isolated and the extracted RNA was subjected to next-generation sequencing. We performed directional RNA sequencing for these samples and subsequently quantified gene expression levels. Sequenced samples depicted high expression levels for known hippocampal markers such as *Gad1*, *Gad2*, *Slc17a7*, and *Mbp* (*Figure 2—figure supplement 1*).

In accordance with increased cell death during the active neurodegeneration phase at 6 months, levels of *Trp53* were elevated in APP mice of this age compared to controls of the same age (t-test, p<0.05, n = 3/group) (*Figure 2C*). Differential expression analysis at this time point revealed a number of genes that are upregulated in 6-month-old APP mice (*Supplementary file 1*), including 72 B2-SRGs among them (*Supplementary file 1*), of which 13 are learning associated (*Supplementary file 1*). These 72 B2-SRGs were enriched in neural-function-related terms, such as neural development, learning, synapse function, and calcium signaling (*Supplementary file 1*). Consistent with a hypothesis of a transcriptome-wide deregulation that involves B2-SRGs, levels of the 72 B2-SRGs were found to be strongly upregulated during the active phase of neurodegeneration at 6 months (*Figure 2D–F*). In particular, in healthy hippocampi (WT control animals), levels of these genes are normally downregulated in 6-month and 12-month-old mice compared to 3-month-old mice (*p=0.02 and p=0.05, respectively*) (*Figure 2E*). This is in accordance with the higher level of neural synaptic activity and plasticity that younger mice have (*Lilja et al., 2013*). In contrast, levels of B2-SRGs in APP mice remain abnormally high in 6-month-old APP mice compared to the 6-month-old control mice (p<0.05) (*Figure 2E and F*). A validation of our RNA-seq data through RT-qPCR for selected genes that we identified by RNA-seq to be upregulated in 6-month-old APP mice, is presented in *Figure 2—figure supplement 2*. In *Figure 2*, we only tested the expression of those B2-SRGs that overlapped with the upregulated genes revealed by DESeq in 6-month APP mice. When expression dynamics for all B2-SRGs are tested (*Figure 2—figure supplement 3*), these are similar to the ones observed for genes in *Figure 2*, while such dynamics are not observed when a random set of non-B2 RNA regulated genes is tested (*Figure 2—figure supplement 3*). Thus, as explained in *Figure 2—figure supplement 3*, it cannot be excluded that our findings may extend to additional B2-SRGs beyond the 72 identified through our differential gene expression analysis.

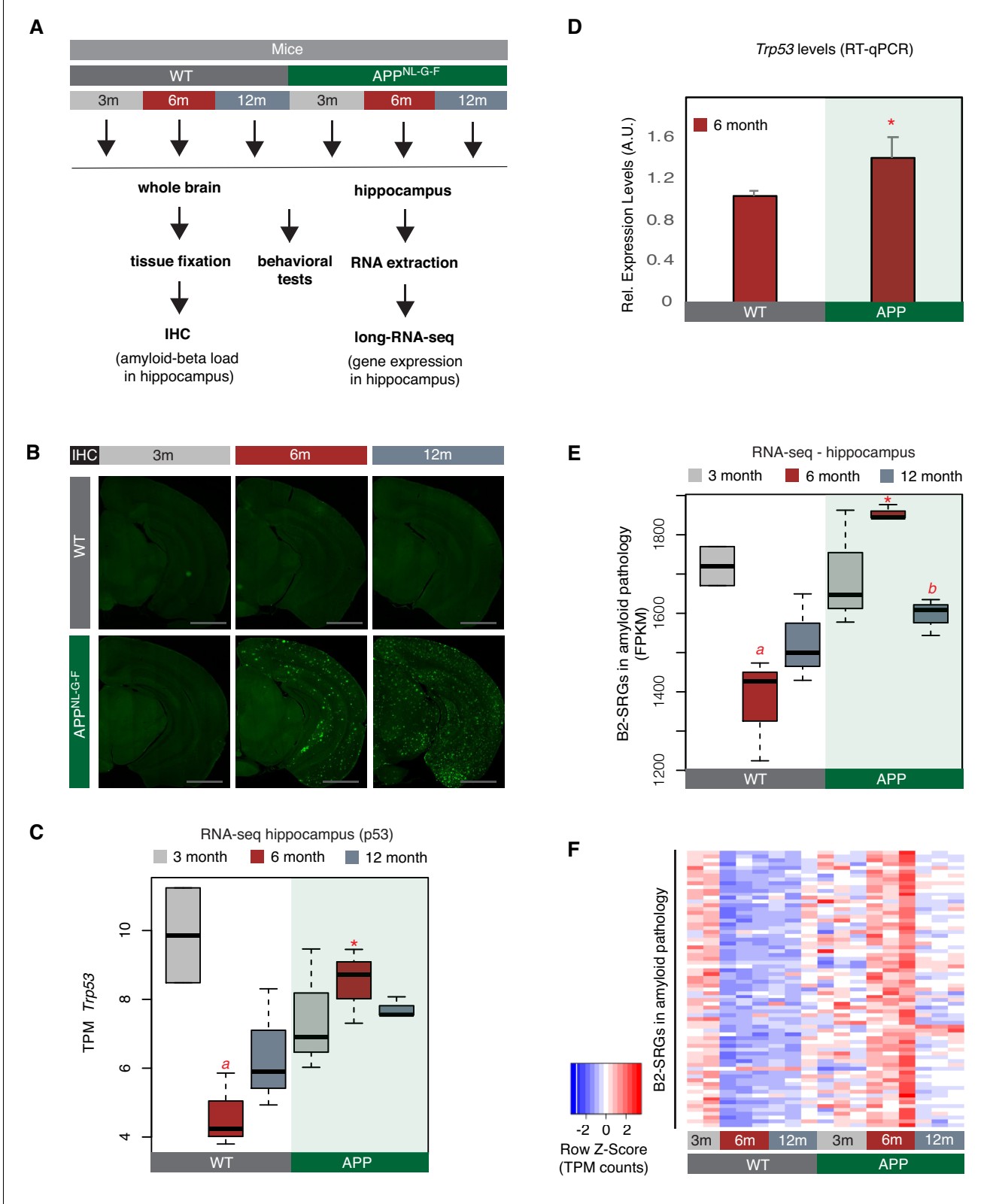

**Figure 2.** A number of B2-SRGs are hyper-activated in amyloid pathology. (A) Experimental design for study of B2-SRGs in the hippocampus of the amyloid pathology mouse model (APP) and the respective wild type (WT) control. (B) Immunohistochemistry for identifying amyloid-beta load in the brain sections of mice from the same cohort as in (A) and our previous study (*Mehla et al., 2019*). Higher fluorescence intensity corresponds to higher amyloid load in APP mice from 6-month-old onwards. (C) Expression levels of the pro-apoptotic gene *Trp53* (official symbol *Trp53*) as defined by long-
*Figure 2 continued on next page*

*Figure 2 continued*

RNA-seq. *Trp53* transcripts per million reads (TPM) from the APP mice compared to WT, grouped by age. Boxplot depicts distribution of expression levels of *Trp53* gene among different age groups of mice (WT: wild type, APP: mice with amyloid pathology). Black line denotes median. Statistical significance (p value threshold 0.05) for the comparison between 6-month-old APP and 6-month-old WT (p=0.01) (depicted as asterisk, unpaired, non-directional t-test, n = 3/group) and the comparison between 3-month-old WT (n = 2) and 6-month-old WT (n = 3) (p=0.04) (depicted as *a*). No significance for the rest of the comparisons between APP and WT in the other two age groups, or between other different ages within the same group (APP or WT). (D) Expression levels of the pro-apoptotic gene *Trp53* (official symbol *Trp53*) in 6-month-old mice as defined by RT-qPCR. Statistical significance (p value threshold 0.05) for 6-month-old APP greater than 6-month-old WT(depicted as asterisk, unpaired directional t-test, n = 3/group, error bars represent standard deviation from the mean). (E) Expression levels as defined by long-RNA-seq of B2-SRGs that are upregulated in amyloid pathology (72 genes, *Supplementary file 1*). Boxplot depicts distribution of expression levels (in FPKM/Fragments per Kilobase per Million) of B2-SRGs among different age groups of wild type and APP mice. Statistical significance (p value threshold 0.05) for the comparison between 6-month-old APP and 6-month-old WT (p=0.02) (depicted as asterisk, unpaired, non-directional t-test, n = 3/group), for the comparison between 3-month-old WT (n = 2) and 6-month-old WT (n = 3) (p=0.03) (depicted as *a*) and for the comparison between 6-month-old APP and 12-month-old APP (p=0.005) (depicted as *b*). No significance for the rest of the comparisons between APP and WT in the other two age groups, or between other different ages within the same group (APP or WT). (F) Gene expression levels of B2-SRGs that are upregulated in amyloid pathology (72 genes, *Supplementary file 1*) show strong association with amyloid pathology status in the hippocampus of APP and WT mice of different ages. Heatmap depicts gene expression with rows representing B2 RNA regulated genes and columns representing mouse samples. Expression values are normalized per row and correspond to TPM values. Red color represents higher expression.

The online version of this article includes the following figure supplement(s) for figure 2:

**Figure supplement 1.** Expression of known hippocampal markers in our RNA-seq data of mice hippocampi.
**Figure supplement 2.** Validation of RNA-seq data in WT and APP mice by RT-qPCR.
**Figure supplement 3.** Expression levels of all B2-SRGs in amyloid pathology.

These findings show a transcriptome wide hyper-activation of certain B2-SRGs during the active neurodegeneration phase of amyloid pathology.

## B2 RNA processing ratio increases during the neurodegeneration phase of in vivo amyloid pathology

Given the role of B2 RNAs in the regulation of SRGs, we hypothesized that the observed hyperactivation of B2-SRGs in the hippocampus of 6-month-old APP mice may reflect a similar upstream deregulation at the level of the B2 RNAs.

To test this, we employed a customized version of RNA sequencing and analysis used in our previous study (*Zovoilis et al., 2016*) that allows for enrichment and sequencing of the short SINE RNA fragments (<100 nt) produced by B2 RNA processing (short-RNA-seq). In contrast to standard long-RNA-seq protocols that include RNA fragmentation and, thus, may introduce bias, this approach circumvents this problem. Moreover, the long-RNA-seq protocols exclude short RNA fragments of <100 nt making the identification of B2 short fragments challenging. After short-RNA-seq (*Figure 3A*), mapping of the 5′ ends of the sequenced fragments across the B2 loci enables the determination of processing points at B2 RNA (depicted as 'X' in *Figure 3A*), including those at the critical RNA Pol II binding region (depicted as a rectangle in *Figure 3A*). *Figure 3B* depicts an example of these fragments from one of the samples. As shown in *Figure 3A*, B2 RNA is extensively processed in hippocampi with amyloid pathology. In particular, this data revealed a increased number of B2 RNA fragments in 6-month-old mice compared to controls (*Figure 3A*), a difference that cannot be explained by full-length B2 RNA levels between these groups of mice using long-RNA-seq (*Figure 3C*).

This data suggests that the processing ratio of B2 RNAs may be higher in APP mice of this age. To estimate this ratio, we used the short RNA-seq data in combination with standard long-RNA-seq. As in our previous study, for normalization of short RNA 5′ end values we used a class of short RNAs that is not affected by B2 RNAs, the RNA Pol III-transcribed tRNAs and estimated the levels of B2 RNA processing fragments (*Figure 3E*). Moreover, the absolute numbers of fragmented B2 RNA may vary according to the underlying expression of full-length B2 RNA transcripts. Therefore, in order to factor in any differences in the amount of fragments due to basal expression levels of the full-length B2 RNA, full-length B2 RNA levels were calculated by the directional long-RNA-seq (*Figure 3F*). The long-RNA-seq approach excludes short fragments, and subsequently, B2 RNA fragment values from short RNA-seq were normalized to the levels of the full-length B2 RNAs to calculate the processing ratio. Consistent with our hypothesis and *Figure 3A* findings, 6-month-old APP

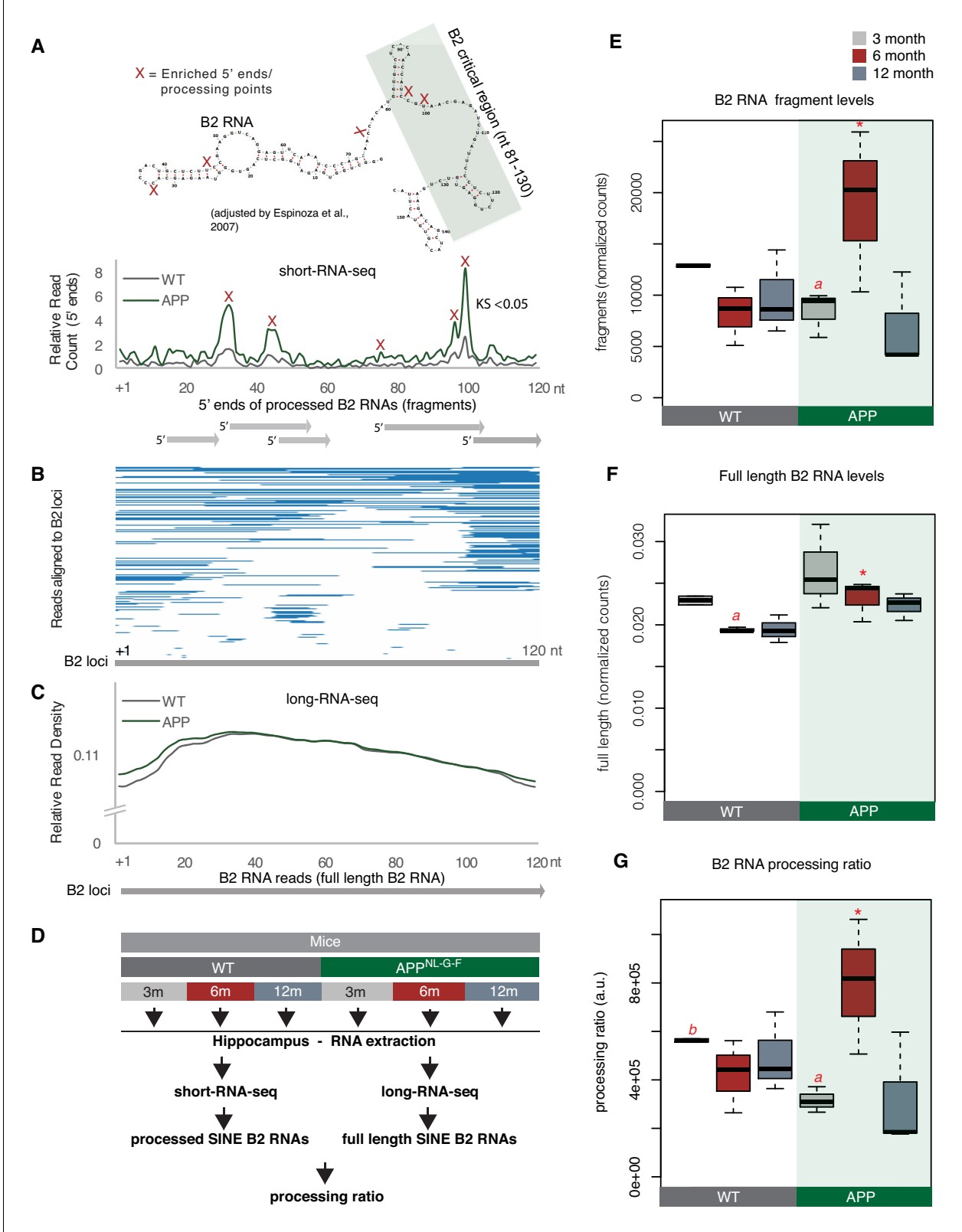

**Figure 3.** B2 RNA processing ratio is increased in 6-month-old APP mice. (**A**) Plotting of the position of the first base (5' end) of B2 RNA fragments across the B2 loci to depict increased levels of B2 RNA fragments in 6-month-old APP mice. *Upper panel*: Secondary structure and processing points of B2 RNA. Secondary structure of B2 RNA adapted from Espinosa and colleagues (*Espinoza et al., 2007*). As in our previous study (*Zovoilis et al., 2016*), we depict the B2 RNA processing points based on short RNA-seq data and mapping of the 5' ends of B2 RNA fragments. X marks which

*Figure 3 continued on next page*

*Figure 3 continued*

cleavage sites of B2 RNA in the upper panel correspond to enriched processing points (the peaks of 5′ end fragments distribution) at the lower panel. The green rectangle depicts the critical region that binds and suppresses RNA Pol II (*Yakovchuk et al., 2009*; *Espinoza et al., 2007*; *Ponicsan et al., 2010*; *Ponicsan et al., 2015*) that may be affected by such processing points. *Lower panel*: Distribution of the 5′ends of B2 RNA fragments across the B2 RNA loci based on mapped short RNA-seq from the hippocampi of 6-month-old APP and WT mice. The x axis represents a metagene combining all B2 RNA loci aligned at the start site of their consensus sequence (position +1) and the y axis shows the relative 5′ end count for B2 RNA fragments aligning to any position downstream of position +1. Figure depicts the difference between B2 RNA fragment distribution of 6-month-old APP mice (higher peaks) vs 6-month-old WT mice (lower peaks) (Kolmogorov Smirnov test, KS <0.05). (B) Alignment of short-RNA seq reads mapped against the multiple B2 loci. Only B2 loci with at least one read are depicted. Short RNA-seq reads from one of the 6-month-old APP mice, whose 5′ ends are plotted at (A) lower panel are used here as an example of the length and position of fragments across B2 loci. (C) Metagene analysis of long-RNA-seq reads of mice across B2 RNA loci that is enriched in full-length B2 RNAs but not in B2 RNA fragments. Comparative analysis shows low difference in relative read density of full-length B2 RNAs between WT and 6-month-old APP. X axis as in A. Y axis corresponds to read coverage across the B2 loci. (D) Experimental design for estimation of B2 RNA processing ratio based on short and long-RNA seq data for all APP mice and controls. (E) Boxplot depicts distribution of levels of processed SINE B2 RNA fragments among different age groups of mice between wild type and APP. Statistical significance (p value threshold 0.05) only for 6-month-old APP greater than 3-month-old APP (p=0.04), 12-month-old APP (p=0.04) and 6-month-old WT (p=0.04) (depicted as asterisk, n = 3/group, unpaired directional t-test). All the other comparisons between groups and different ages non-significant except 3-month-old WT less than 3-month-old APP (p=0.03)(depicted as a). (F) Boxplot depicts distribution of levels of full-length SINE B2 RNAs among different age groups of mice between wild type and APP. Statistical significance (p value threshold 0.05) for the comparison between 6-month-old APP and 6-month-old WT (p=0.05; depicted as asterisk, n = 3/group, non-directional unpaired t-test) and for the comparison between 3-month-old WT (n = 2) and 6-month-old WT (n = 3) (p=0.004; depicted as a). All the other comparisons between groups and different ages are non-significant. (G) Boxplot depicts distribution of SINE B2 RNA processing ratio among different age groups of mice between wild type and APP. Statistical significance (p value threshold 0.05) for the comparison between 3-month APP and 6-month-old APP (p=0.04; depicted as *a*, n = 3/group, non-directional unpaired t-test), for the comparison between 3-month-old WT (n = 2) and 3-month-old APP (n = 3) (p=0.008; depicted as *b*), and for 6-month-old APP greater than 6-month-old WT (p=0.05; depicted as asterisk, n = 3/group). All the other comparisons between groups and different ages are non-significant.

The online version of this article includes the following figure supplement(s) for figure 3:

**Figure supplement 1.** Plotting of the position of the first base (5′ end) of B2 RNA fragments across the B2 loci to compare levels of B2 RNA fragments between APP and WT mice in the three different age groups.

**Figure supplement 2.** Plotting of the position of the first base (5′ end) of B2 RNA fragments across the B2 loci to compare levels of B2 RNA fragments in 6-month-old mice between B2 elements that overlap exonic/genic regions and those that do not.

---

mice were found to have substantially increased ratio of B2 RNA processing compared to control mice of the same age (*Figure 3G*) (p<*0.05*, n = 3/group). This increase was observed only in the 6-month-old mice (*Figure 3G*, *Figure 3—figure supplement 2*) and coincides with the increase in *Trp53* levels and B2-SRGs levels observed in this age.

Thus, APP mice at the active neurodegeneration phase are characterized by higher destabilization and processing ratio of B2 RNAs, consistent with the observed increase in B2-SRG levels in these same animals.

## Hsf1 accelerates B2 RNA processing

We then focused on the molecular mechanism underlying the increased B2 RNA processing during response to amyloid toxicity. When we had previously examined the mechanism of B2 RNA processing in non-neural cells (NIH/3T3 cells during heat shock), a member of the PRC2 protein complex, Ezh2, was reported as being responsible for the B2-RNA-accelerated destabilization and processing during response to stress (*Zovoilis et al., 2016*). However, as shown in *Figure 4—figure supplement 1A–B*, scant expression levels of Ezh2 levels in neural cells indicate that Ezh2 is not a key factor in B2 RNA processing in brain. Thus, the factors that may mediate destabilization for B2 RNAs during stress in neural cells remain elusive.

In our earlier study that described the induction of B2 RNA processing by Ezh2, it remained unclear how Ezh2 exerted its impact on B2 RNAs, since Ezh2 lacked any known RNAse activity (*Zovoilis et al., 2016*). However, in subsequent experiments (*Hernandez et al., 2020*), we showed that instability is in fact inherent to the B2 RNA molecule while interaction with Ezh2 only accelerates this destabilization and Ezh2 does not cleave B2 RNA by itself. This finding suggests that other proteins may have a similar effect on B2 RNA stability. Therefore, we started searching for stress-related candidate proteins that could affect the B2 RNA processing.

We showed before that, during response to stress in NIH/3T3 cells, B2 RNA binding is enriched near stalled RNA polymerase genomic sites. These areas are known to be highly enriched in binding

sites of various stress related proteins, among which Hsf1, a master regulator of stress response for various types of cellular stress (*Pandey et al., 2011*). Hsf1 has been previously connected with activation of SRGs through both transcriptional factor (TF) activities as well as other yet unknown TF-independent processes (*Inouye et al., 2007*). Interestingly, when we examined the proximity of Hsf1 binding sites, identified by the Lis lab (*Mahat et al., 2016*), to genes with B2 RNA binding sites (*Supplementary file 1*; *Zovoilis et al., 2016*), we found that increased number of Hsf1-binding sites were found near B2-SRGs (*Figure 4—figure supplement 1C*) and were further enriched after application of a heat-shock stimulus (KS-test <0.05). Moreover, as shown in *Figure 4—figure supplement 1A–B*, in contrast to *Ezh2*, *Hsf1* is expressed in neural tissues and especially in hippocampus. These findings provided an indication that *Hsf1* may be implicated in B2 RNA biology and that it is also expressed in neural cells, which urged us to test further its expression pattern in our biological context. *Hsf1* levels were found to be upregulated in APP 6-month-old mice compared to control group of the same age (p<*0.05*, n = 3/group) (*Figure 4A*) in the RNA-seq data, a result confirmed also through RT-qPCR (*Figure 4B*). As mentioned above, at the same time, 6-month-old APP mice have increased B2 RNA processing ratio. Thus, this data suggests that Hsf1 may be a good candidate for accelerating B2 RNA processing in the context of amyloid pathology.

To check this, we investigated whether the interaction between B2 RNA and Hsf1 can accelerate B2 RNA destabilization. In particular, we incubated full-length B2 RNA with the Hsf1 protein in vitro. As shown in *Figure 5A*, in the presence of Hsf1, the destabilization of full-length B2 RNA was accelerated in contrast to the control protein (PNK) or in the absence of protein (no protein). No processing was observed in an RNA fragment co-incubated with B2 RNA and the protein (marked with an asterisk across *Figure 5*). We then questioned whether the fragments observed during B2 RNA processing in vitro are the same with those we observed in NIH/3T3 cells in our previous study and hippocampal cells in our current study (*Figure 3A*). To test this, we subjected the in vitro processed B2 RNA fragments to short RNA-sequencing as with the in vivo samples and compared the processing patterns between the two cases. As shown in *Figure 5—figure supplement 1A*, the in vitro processing points were the same for most of the positions identified in vivo including the most prominent

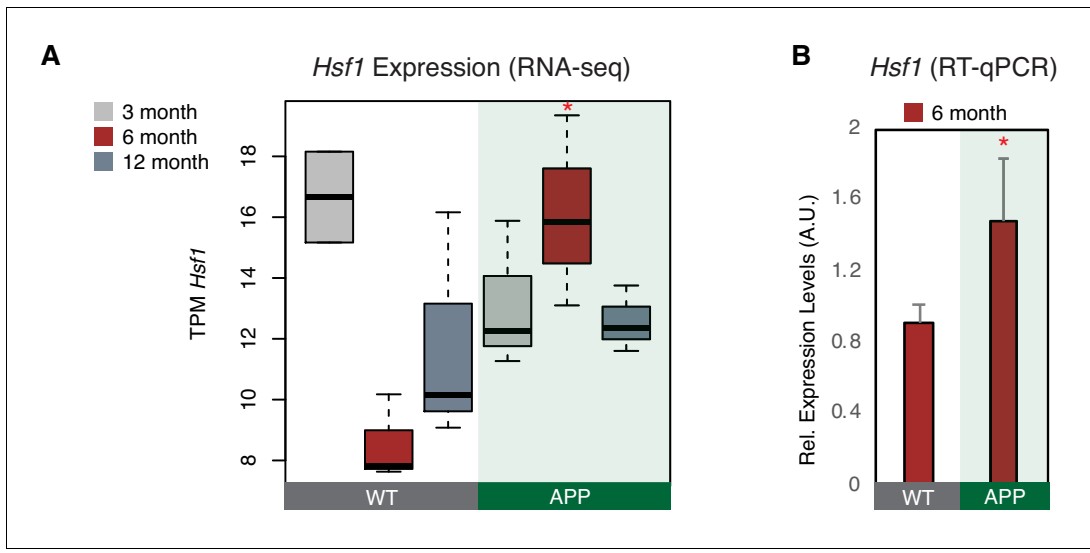

**Figure 4.** *Hsf1* is upregulated in 6-month-old APP mice. (**A**) Boxplot depicts distribution of expression levels of *Hsf1* gene among different age groups of mice between wild type and APP. Values are based on TPM counts of long-RNA-seq data. Statistical significance (p value threshold 0.05) for the comparison between 6-month-old APP and 6-month-old WT (p=0.03) (n = 3/group, unpaired non-directional t-test). All the other comparisons between groups and different ages are non-significant. (**B**) Expression levels of *Hsf1* in 6-month-old mice as defined by RT-qPCR. Statistical significance (p value threshold 0.05) for 6-month-old APP greater than 6-month-old WT (depicted as asterisk, unpaired directional t-test, n = 3/group, error bars represent standard deviation from the mean). The online version of this article includes the following figure supplement(s) for figure 4:

**Figure supplement 1.** Expression of *Hsf1* in neural tissues.

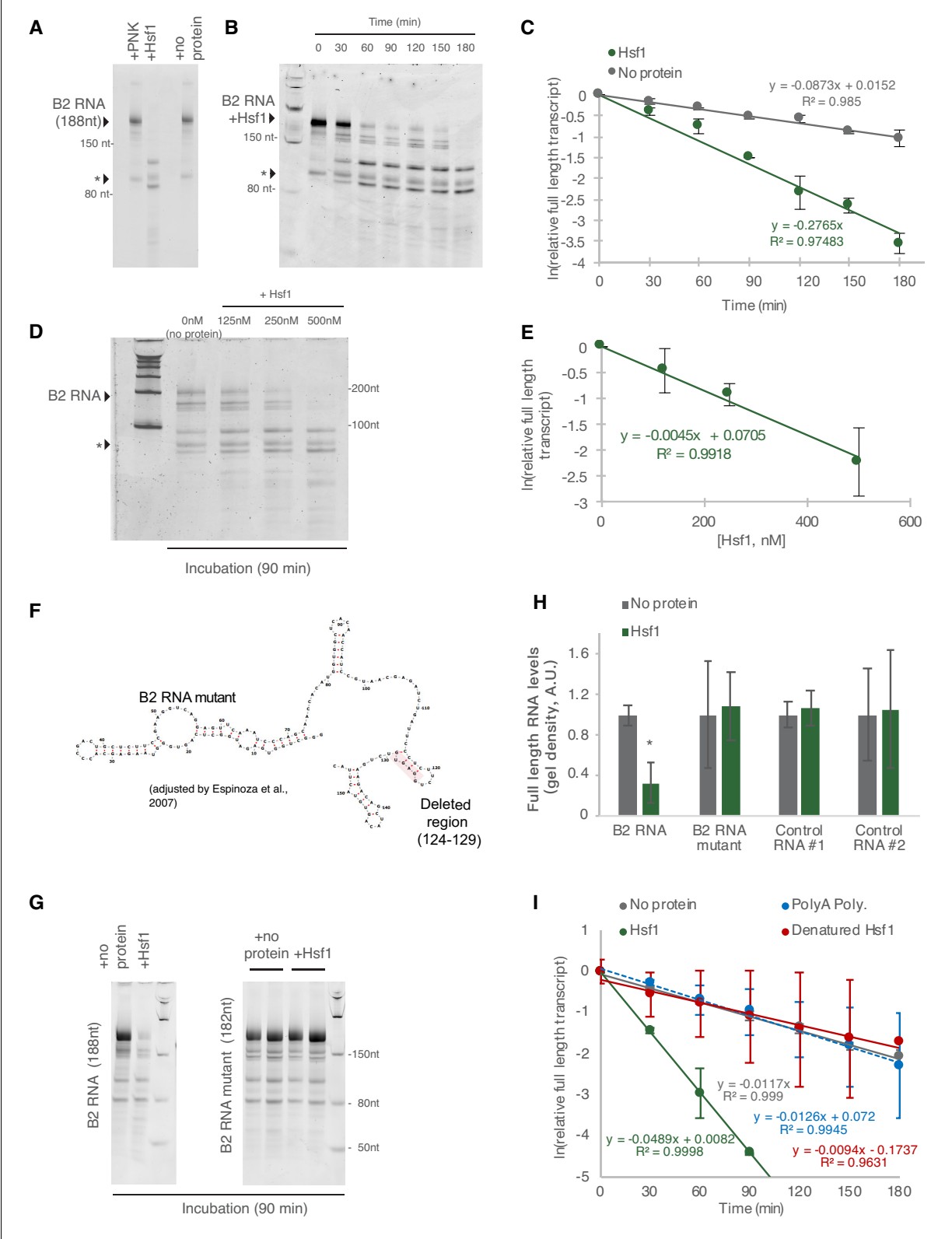

**Figure 5.** Hsf1 accelerates B2 RNA processing. (A) In vitro incubation of B2 RNA. In vitro transcribed and folded B2 RNA at 200 nM incubated with PNK as a control (lane 1), 250 nM Hsf1 (lane 2) and without protein (lane 4). Incubations occurred for 6 hr at 37°C. (B) In vitro incubation of B2 RNA for different incubation periods. In vitro transcribed B2 RNA (100 nM) incubated at 37°C with 500 nM Hsf1 in the course of 180 min with time intervals of 30 min. (C) Relative full-length RNA remaining from (B) using ImageJ area under the curve software over time. The full gels are available as source files. (D)

*Figure 5 continued on next page*

*Figure 5 continued*

Titration of Hsf1 protein in incubation with 200 nM in vitro transcribed B2 RNA. Concentrations of Hsf1 range from 0 to 500 nM (0 means incubation was performed in the absence of protein and only in the presence of TAP buffer, for the same duration as the other three conditions). (E) Relative full-length RNA remaining from (D) using ImageJ area under the curve software over time. The trendline is a linear fit, displaying standard deviation on the data points. The full gels are available as source files. (F) Graphical representation of the deletion in the B2 RNA mutant on the secondary structure of B2 RNA as adapted from *Espinoza et al., 2007*. (G) In vitro incubation of the B2 RNA and the B2 RNA mutant for 90 min at 37°C with either Hsf1 or no protein (just TAP buffer, see Materials and methods). (H) Comparison among B2 RNA, the B2 RNA mutant (6nt deletion) and two control RNAs regarding the full-length RNA levels remaining after in vitro incubation for 90 min at 37°C with Hsf1. Sizes of control RNAs are control for RNA #1, 143nt and for control RNA #2, 163nt. or no protein (just TAP buffer, see Materials and methods). Incubation in the absence of Hsf1 but presence of the same buffer (TAP) was used as control to take into account any non-Hsf1-specific RNA destabilization due to non-specific degradation. Asterisk represents statistical significance p=0.005 (unpaired, non-directional t-test) for the comparison between Hsf1 and no protein incubation (three replicates for B2 RNA). The full gels are available as source files. (I) Comparison among Hsf1 (~60 KDa), denatured Hsf1, Poly A polymerase (~56 KDa) and no protein (just TAP buffer) with regard to B2 RNA processing (estimating relative full-length RNA remaining as in C). The full gels are available as source files. For Hsf1, after 90 min no detectable full-length B2 RNA could be measured anymore and no further data points were included in the linear model.

The online version of this article includes the following source data and figure supplement(s) for figure 5:

**Source data 1.** Full gel images for *Figure 5*, part 1.

**Source data 2.** Full gel images for *Figure 5*, part 2.

**Figure supplement 1.** Position of B2 RNA fragments generated in vitro and downregulation of Hsf1 protein levels.

ones around 99 and 33. The two processing patterns differed only at positions 90 and 47, suggesting that in vivo B2 RNAs may be protected or processed in these two positions by yet unidentified mechanisms.

*Figure 5B* shows B2 RNA destabilization in time. B2 RNA destabilization in the presence of Hsf1 was accelerated compared to no protein (*Figure 5C* and *Figure 5—source datas 1* and *2*). Consistent with the above finding, the rate of processing of B2 RNA was dependent on Hsf1 concentration in the reaction (*Figure 5D and E*) and increased upon increase of Hsf1 concentration. In *Figure 5D* and our analysis in *Figure 5E*, as a zero concentration point (no Hsf1 protein) we used RNA incubated in the same buffer and for the same time as the samples with the Hsf1 protein to take into account any RNA destabilization due to non-specific degradation, hydrolysis or B2 RNA endogenous self-cleavage. In order to confirm further the Hsf1-mediated acceleration of B2 RNA we transcribed a mutant B2 RNA that harbors a 6nt deletion in the hairpin at position 124 (*Figure 5F*). Then, we compared B2 RNA processing during incubation in the presence and absence of Hsf1 and observed an impairment in the ability of Hsf1 to accelerate B2 RNA processing (*Figure 5G–H*). Similar negative results were obtained when other control RNAs were tested (*Figure 5H*, control RNA #1, 143nt; control RNA #2, 163nt). Subsequently, we tested the impact on acceleration of B2 RNA processing by Hsf1 in the case of heat denaturation of Hsf1. As shown in *Figure 5I*, heat denaturation of Hsf1 resulted in abrogation of its ability to accelerate B2 RNA processing (*Figure 5I*). Similar results were obtained with the incubation of B2 RNA with another RNA binding protein of similar size to Hsf1 (Hsf1 ~60 KDa, Poly A polymerase ~56 KDa) (*Figure 5I*).

This data shows that Hsf1 has the potential to accelerate B2 RNA processing in vitro.

## Hsf1 mediates increased B2 RNA processing in response to amyloid beta toxicity

The increased levels of Hsf1/B2 RNA processing in APP 6-month-old mice raised the question whether response to amyloid toxicity in hippocampal cells is connected with an Hsf1-mediated increase in B2 RNA processing. In order to test this, we employed a hippocampal cell culture model using the a HT-22 cell line, which has been used extensively in the past as hippocampal cell stress model (*Davis and Maher, 1994*; *Liu et al., 2009*). We incubated these cells with amyloid beta peptides (1–42 aa) and compared their transcriptome to the one of cells incubated with an inverted sequence control peptide (R, reverse 42–1) (*Figure 6A*). Incubation of these cells with 1–42 amyloid beta peptides results in upregulation of a number of genes (*Supplementary file 1*), 25 of which are also found upregulated in amyloid beta pathology (6-month-old APP mice) (*Supplementary file 1*). The increase in expression levels of genes associated with amyloid pathology (*Figure 6B and C*) suggests that this cellular model simulates to a certain extend the amyloid toxicity effect on the

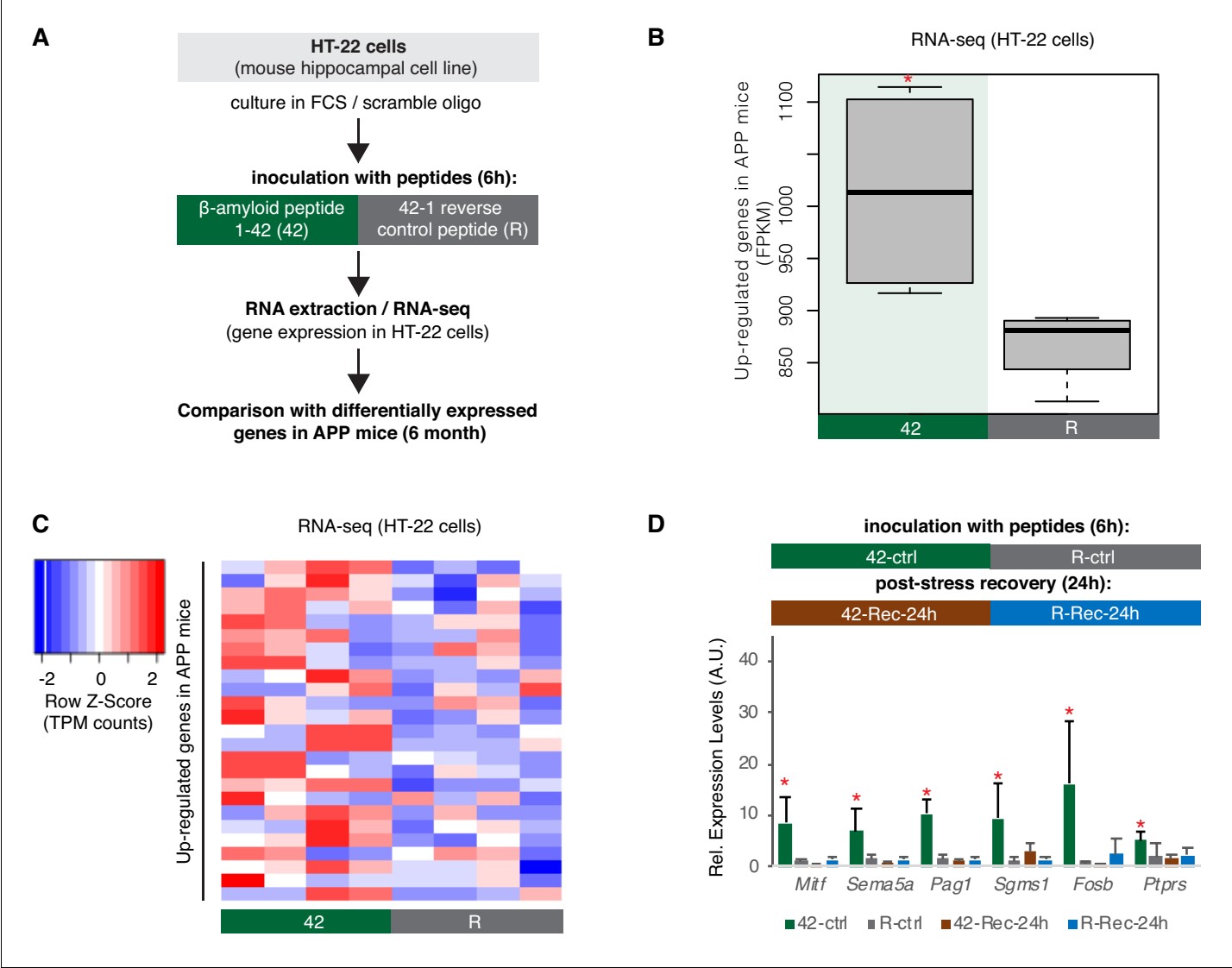

**Figure 6.** A hippocampal cell culture assay for tracking effects of amyloid beta toxicity on B2 RNA stability. (**A**) Experimental design for the amyloid toxicity cell culture assay employing HT-22 cells. Cells culture media were supplemented with Fetal Calf Serum (FCS) and the scramble LNA described in *Figure 8* to allow comparison with the LNA experiments. (**B**) Expression levels as defined by long-RNA-seq of genes upregulated in amyloid pathology (6-month-old APP mice) that are also upregulated in amyloid toxicity (25 genes, *Supplementary file 1*). Boxplot depicts distribution of expression levels (in FPKM/Fragments per Kilobase per Million) of these genes between the two conditions in HT-22 cells. p=0.05 for the comparison between HT22 samples transfected with 1–42 (*Espinoza et al., 2007*) vs 42–1 reverse control peptide (**R**) (depicted as asterisk, unpaired non-directional t-test, n = 4/group). (**C**) Heat map showing gene expression changes during incubation with amyloid peptides in HT22 cells for genes that are also upregulated in amyloid pathology (25 genes, *Supplementary file 1*) to test whether our cell culture system can simulate the transcriptome observed in amyloid pathology in vivo. Heatmap depicts increase in expression values calculated by long-RNA-seq (TPM values) in the HT22 samples transfected with 1–42 (*Espinoza et al., 2007*) vs 42–1 reverse control peptide (**R**) (heatmap columns), for these genes (*Supplementary file 1*) (heatmap rows). TPM values are normalized per row. Red color represents higher expression. (**D**) Upper panel: Experimental design for the amyloid toxicity assay and a control experiment testing recovery 24 hr after application of the stress stimulus to confirm return of gene expression levels to pre-stress levels. Lower panel: Confirmation through RT-qPCR of the expression differences of selected B2-SRGs identified in RNA-seq data to differ between HT22 cells transfected with 1–42 (*Espinoza et al., 2007*) and cells transfected with the 42–1 reverse control peptide (**R**) for 6 hr (*Supplementary file 1*). Statistical significance (p value threshold 0.05) is depicted as asterisk for expression in treated cells (*Espinoza et al., 2007*) greater than the expression in controls (**R**) (unpaired directional t-test, n = 4, error bars represent standard deviation from the mean). In the same graph, we include also expression levels of cells from the same experiment that were left to recover in standard medium without any peptides for 24 hr after the initial 6 hr application of the peptides (42-Rec and R-Rec). Expression levels in all genes tested returned to pre-stress levels. Asterisk depicts statistical significance as described above also for the comparison between 42-ctrl and 42-Rec-24h.

The online version of this article includes the following figure supplement(s) for figure 6:

*Figure 6 continued on next page*

*Figure 6 continued*

**Figure supplement 1.** PCA plots and correlation matrix for sequenced samples in amyloid pathology and amyloid toxicity models.

transcriptome of hippocampal cells and their response to cellular stress. However, as any cell culture system, it has limitations regarding tissue level functions such as learning.

Genes upregulated in our amyloid toxicity model included 25 B2-SRGs (*Supplementary file 1*). When testing for enriched terms in these 25 genes, biological processes related to apoptosis, such as regulation of apoptotic process and programmed cell death were at the top of the list (*Supplementary file 1*) and included, among others, genes such as *Fosb* and *Mitf* that have been connected with AD (*Solé-Domènech et al., 2016*; *Gupta et al., 1986*). Validation of our RNA-seq data in HT-22 cells through RT-qPCR for selected B2-SRGs identified to be upregulated in 42 vs. R is presented in *Figure 6D*. In the same figure, we also present the expression levels of these genes during recovery, 24 hr after application of the amyloid beta toxicity stimulus, confirming the return to pre-stimulus levels and the specificity of the treatment with amyloid beta peptides (*Figure 6D*).

Out of the 25 genes that are upregulated in both mice and our cell culture system, six are B2-SRGs (*4932438A13Rik*, *Fosb*, *Pag1*, *Ptprs*, *Sema5a*, and *Sgms1*) and include a well-known immediate early gene (*Fosb*), genes associated with sensitivity to amyloid toxicity (*Pag1*, *Sema5a*, *Sgms1*, *Fosb*) (*Hadar et al., 2016*; *Lin et al., 2009*; *Hsiao et al., 2013*), as well as genes associated with *Trp53* (*Ptprs*, *Fosb*)(*Motiwala and Jacob, 2006*; *Liu et al., 2018*).

We then questioned whether by inducing an artificial degradation of B2 RNA, we would be able to induce expression of B2-SRGs in our model system of HT22 cells in the absence of any stimulus such as amyloid beta. To achieve this, we employed a similar approach and the same LNAs against B2 RNA that we used in our previous study (NIH/3T3, heat shock). Application of the LNA against B2 (*Figure 7A*) was able to reduce levels of B2 RNA compared to the control LNA (*Figure 7B*). Similarly to NIH/3T3 cells, targeting of the B2 RNA in HT22 cells, resulted in the increase of the expression levels of selected B2-SRGs that are upregulated in amyloid beta pathology (see *Figure 2— figure supplement 2*, the first 10 genes and *Figure 6D*). The increase in gene expression occurred in the absence of the stress stimulus, in this case amyloid beta, suggesting that these genes are under the suppressive control of B2 RNAs in HT22 cells (*Figure 7C*). At the same time, B2 RNA destabilization did not affect expression of five non-B2-SRGs that were used as negative controls, including example genes, such as Adcy1 and Kalrn, that are nevertheless upregulated in amyloid beta pathology (*Figure 2—figure supplement 2*, the last two genes).

Subsequently, in order to test the impact of amyloid toxicity to Hsf1-mediated B2 RNA processing we treated HT22 cells with either an LNA against *Hsf1* or a scramble LNA (control) followed by incubation with the 1–42 peptides, that subject the cells to amyloid toxicity stress, or the respective control peptide (*Figure 8A*, same experimental design and cells as in *Figure 6*). As shown in *Figure 8B*, treatment with the anti-*Hsf1* LNA suppressed any increase in *Hsf1* levels between 42-anti-*Hsf1* and R-anti-*Hsf1* cells, while this increase is observed between non-*Hsf1* LNA-treated cells upon application of the amyloid beta peptides (42-ctrl vs R-ctrl). Treatment with with the anti-*Hsf1* LNA during transfection with the amyloid beta peptides also led to a decrease in protein levels of *Hsf1* (*Figure 5—figure supplement 1B*). As observed in the APP mice, amyloid beta peptides resulted in increased B2 RNA processing only in cells without *Hsf1* knock down, suggesting that it is the toxicity of these peptides that induces the SINE B2 RNA transcriptome changes (*Figure 8C*, *Figure 8—figure supplement 1*). In contrast, under anti-*Hsf1* LNA, application of the 1–42 peptides was unable to increase B2 RNA processing in the cells compared to the control cells (*Figure 8C*). A similar effect was observed in B2-SRGs that are upregulated during amyloid toxicity (*Supplementary file 1*, *Figure 8D–F*). A selected number of genes was also tested with RT-qPCR to confirm the patterns observed in the RNA-seq data (*Figure 8E*). As in case of amyloid pathology, it cannot be excluded that our findings may extend to additional B2-SRGs (*Figure 8—figure supplement 2*).

These results suggest that amyloid beta toxicity induces B2 RNA processing also in vitro and *Hsf1* comprises a necessary component in the upstream activation pathway of B2 RNA processing and, thus, of the genes regulated by B2 RNA.

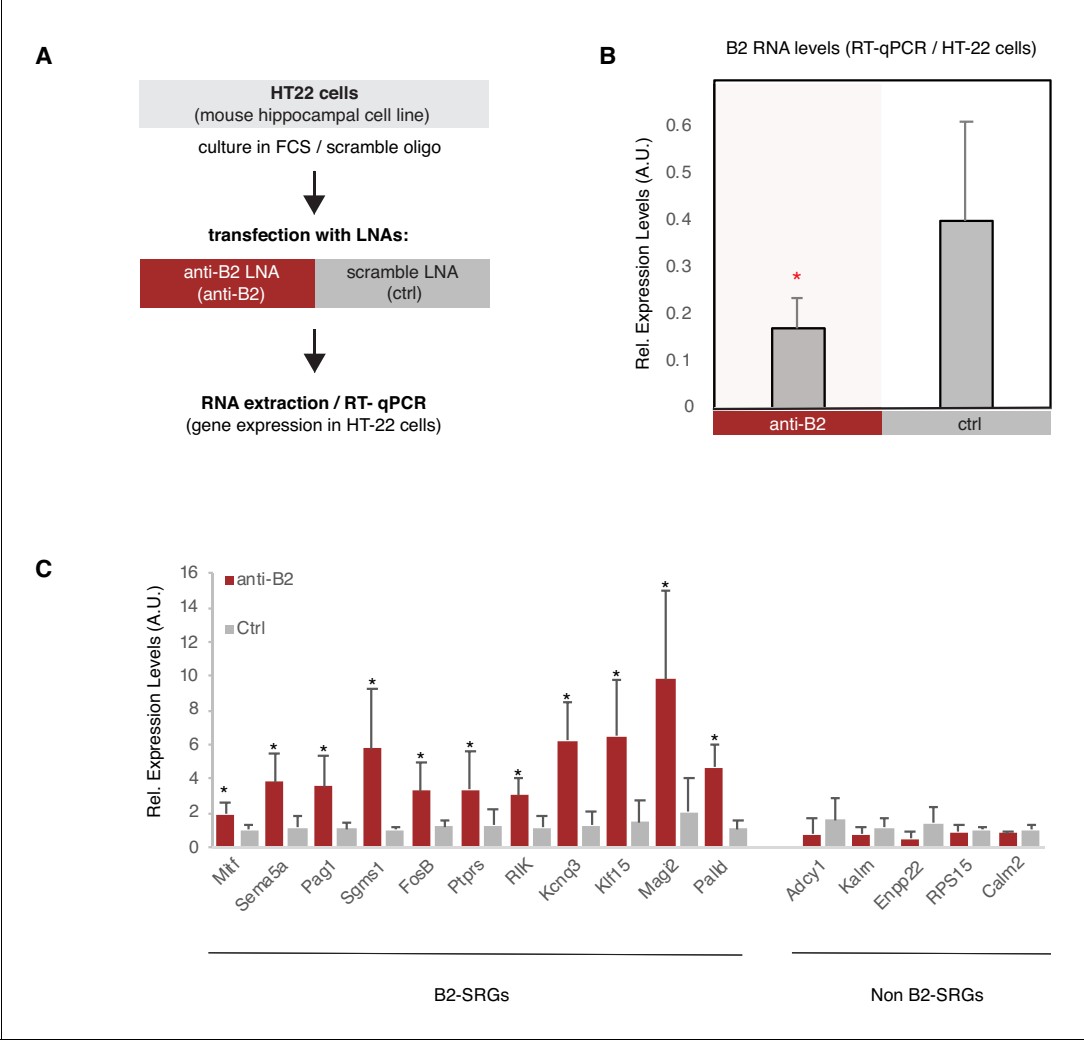

**Figure 7.** B2 NA destabilization leads to increase in expression of B2-SRGs. (**A**) Experimental design for the B2 RNA knock-down cell culture assay employing HT-22 cells. (**B**) Expression levels of full-length B2 RNA (RT-qPCR) in the B2 RNA KD experiment. Statistical significance (p value threshold 0.05) for anti-B2 less than control (depicted as asterisk, n = 3, unpaired directional t-test, error bars represent standard deviation from the mean). (**C**) Expression levels (RT-qPCR)_of selected B2-SRGs (see *Figure 2—figure supplement 2* and *Figure 6D*) and the respective negative controls during B2 RNA KD. Statistical significance (p value threshold 0.05) for anti-B2 greater than control (depicted as asterisk, n = 3/group, unpaired directional t-test, error bars represent standard deviation from the mean). Negative controls include five non-B2-SRGs that show no statistically significant difference between the two conditions.

## Discussion

Fewer than five ribozymes have been identified in mammals, including our previous work on RNAs made from two retrotransposon families, murine SINE B2 RNAs and human SINE ALU RNAs, which are self-cleaving RNAs (*Hernandez et al., 2020*). We previously showed that cleavage in SINE B2 RNAs controls response to cellular stress through activation of stress response genes in heat shock. However, no connection between this novel molecular mechanism and pathologic processes was until now known. Moreover, since B2 RNAs are intrinsically reactive, and contact with Ezh2 only accelerates cleavage, it remained plausible that other stress-related proteins may also have a similar effect on accelerating B2 RNA processing, which would link this ribozyme-like property to stress response through pathways other than Ezh2. This is especially relevant in mouse tissues, such as the brain, where Ezh2 expression is limited, an expression pattern observed also in human (Human Protein Atlas) (*Uhlén et al., 2015*).

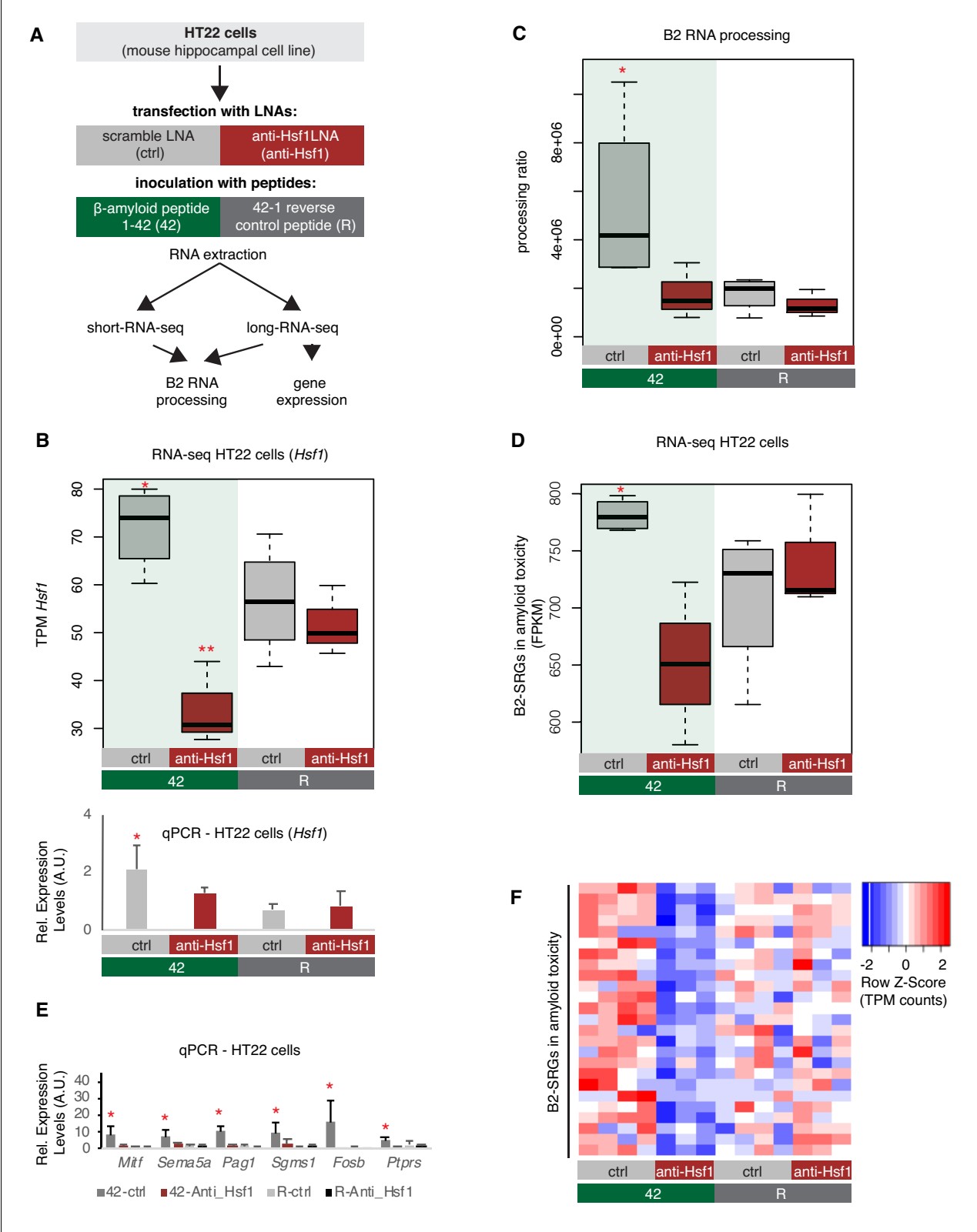

**Figure 8.** Hsf1 mediates B2 RNA processing in amyloid toxicity. (**A**) Experimental design of the combined *Hsf1* Knock Down – amyloid toxicity assay in HT22 cells followed by short and long RNA-seq. (**B**) Expression levels of *Hsf1* as defined by long-RNA-seq (upper panel) and RT-qPCR (lower panel). Boxplots compare *Hsf1* expression (TPM values) during incubation with the scramble LNA, and anti-*Hsf1*-specific LNA, incubated with either the 42 or R amyloid peptides. Statistical significance (p value threshold 0.05) for the comparison between 42/ctrl (n = 4) and 42/anti-*Hsf1* (n = 3), unpaired non-

*Figure 8 continued on next page*

*Figure 8 continued*

directional t-test and for 42/ctrl (n = 4) greater than R/ctrl (n = 4) (both depicted as one asterisk), while two asterisks represent p<0.05 between 42/anti-*Hsf1* and the other three groups. (C) B2 RNA processing ratio based on a combination of short and long-RNA-seq. Boxplot depicts distribution of total SINE B2 RNA processing ratio among different groups of HT22 cells between 42 and R. Statistical significance (p value threshold 0.05) for 42/ctrl greater than R/ctrl (p=0.04, n = 4/group, unpaired directional t-test, depicted as asterisk). No significant difference was observed between 42/anti-*Hsf1* and R/anti-*Hsf1* (n = 3/group) and between R/anti-*Hsf1* and R/ctrl. (D) B2-SRGs expression levels in amyloid beta toxicity based on long-RNA-seq data (FPKM values) for genes of ***Supplementary file 1*** (25 genes). Boxplot depicts distribution of expression levels in HT22 cells between 42 and R. p=0.05 for 42/ctrl greater than R/ctrl (n = 4/group, unpaired directional t-test, depicted as asterisk). No significant difference was observed between 42/anti-*Hsf1* and R/anti-*Hsf1* (NS, n = 3/group) and between R/anti-*Hsf1* and R/ctrl. (E) Expression levels of selected B2-SRGs in the four conditions tested in our amyloid toxicity model though RT-qPCR. Statistical significance (p value threshold 0.05) for 42/ctrl greater than R/ctrl, unpaired directional t-test, n numbers as in subfigure B, with p<0.05 depicted as asterisk (error bars represent standard deviation from the mean). Samples and values depicted for non-*Hsf1* LNA treated samples are the same as in ***Figure 6D*** and are used as controls as these samples were treated with a scramble LNA to allow comparison with the *Hsf1* LNA treated samples. (F) Gene expression levels (long-RNA-seq) of B2-SRGs that are upregulated in amyloid toxicity (***Supplementary file 1***, 25 genes) show strong association with *Hsf1* treatment during response to amyloid toxicity in HT22 cells. Heatmap depicts gene expression with rows as B2-SRGs in amyloid toxicity (***Supplementary file 1***) and columns as different HT22 cell treatments. TPM values are normalized per row. Red color represents higher expression.

The online version of this article includes the following figure supplement(s) for figure 8:

**Figure supplement 1.** B2 RNA levels in HT22 cells and relationship with Hsf1 levels.
**Figure supplement 2.** Expression levels of all B2-SRGs in amyloid beta toxicity.

Here, we unveil increased processing of SINE B2 RNAs as a novel type of transcriptome deregulation underlying amyloid beta neuro-pathology. Our data provides a new link in the murine hippocampal pathways connecting amyloid beta toxicity with transcriptome changes in SRGs through processing of B2 RNAs. In particular, the B2 RNA processing ratio increases upon progression of amyloid pathology both in mouse hippocampus and a hippocampal cell culture model, and B2-SRGs become hyperactivated. Consistent with the spatial proximity between B2-SRGs and Hsf1 binding sites, Hsf1 proved to be key for mediating B2 RNA processing in response to amyloid toxicity. This correlation is observed throughout all sequencing experiments performed in this study (***Figure 8—figure supplement 1***). Our work assigns to *Hsf1* a new function that is independent of its long-established transcription factor function and includes the interaction with and processing of SINE B2 RNAs. The high levels of *Hsf1* trigger a downstream cascade of events which are orchestrated into a cell-wide, SRG-mediated response to stress conditions. This axis is mediated by the ability of B2 RNA to get processed and, thus, act as a molecular switch. Although healthy cells and animals are able to restore the expression levels of *Hsf1*, SRGs and their regulating B2 RNAs upon removal of the stress-generating stimulus, the amyloid beta load in our biological models acts as a continuous stimulus that causes the *Hsf1* - B2 RNA - SRG axis to 'lock' into an activated mode. Upregulation of SRGs results in increased *Trp53* levels that induce neuronal cell death (***Figure 9***).

Stress response genes, that constitute the basis of response to heat shock, have been shown in various studies from us and others to play a critical role also in hippocampal function (***Peleg et al., 2010***; ***Zovoilis et al., 2011***). Thus, our findings here on a potential role of B2 RNAs in the context of neural response to stress constitute a natural continuation of our previous study in heat shock (***Zovoilis et al., 2016***). The fact that B2-SRGs identified in our previous study were found in the current study to be highly enriched in neuronal tissue related biological processes terms and compartments was what has compelled us to investigate further a potential role of B2 RNAs in neural tissue-associated pathologies.

In this study, we employed a mouse model of amyloid pathology in order to test the impact of increased amyloid beta load on B2 RNA processing in vivo. This mouse model has the NL- G-F mutations in the amyloid precursor protein knocked-in to a C57BL genomic background, with each mutation contributing to an increased severity and speed manifestation of the disease (***Mehla et al., 2019***). The combined effect of the APPNL−G−F mutations results in mice that experience rapid onset of AD-like symptoms at approximately 6 months (***Mehla et al., 2019***). This is exactly the time point that we observed the massive hyper-activation of B2 RNA SRGs, *Trp53*, and B2 RNA processing, suggesting that these changes constitute a molecular signature for the active neurodegenerative phase months, before the end state of 12 months that the mice develop terminal AD-like pathology and symptoms.

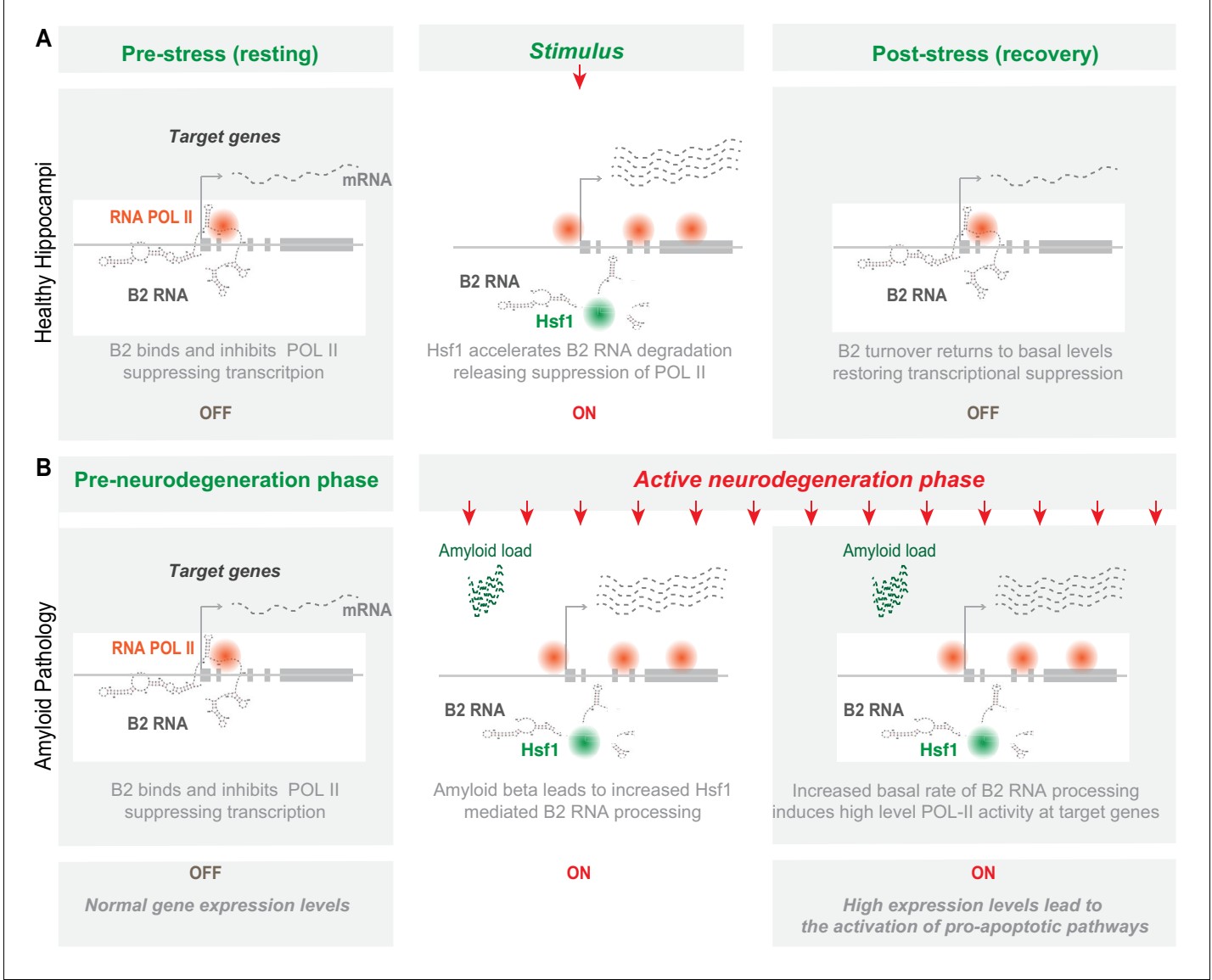

**Figure 9.** Representation of the role of B2 RNA processing in amyloid pathology. Upon removal of the stress-generating stimulus, healthy cells restore the expression levels of *Hsf1*, specific B2 RNA regulated target genes and processing ratio of B2 RNAs returns to base levels. In contrast, in amyloid pathology, increased amyloid beta load acts as a continuous stimulus that causes the *Hsf1* - B2 RNA – B2-SRG axis to 'lock' into an activated mode. ON/OFF represent active and suppressed SRG transcription, respectively.

Interestingly, symptoms and molecular changes are not observed in younger APP mice suggesting the existence of a yet unknown protective mechanism against increased neuronal activity of SRGs in 3-month-old APP mice. This may be attributed to the increased neuronal plasticity observed in younger brains (*Lilja et al., 2013*), which suggest that mechanisms in the younger brain may exist for counter-acting this excessive activity. In particular, stress response genes such as *Fosb* have been shown to participate in specification of cell-type-specific activity-dependent gene programs early in development (*Yap and Greenberg, 2018*). In contrast, during aging, B2-SRG activity in WT mice ramps down, which is not the case in APP mice (*Figure 2*), explaining at the molecular level the increased *Trp53* levels and subsequent activation of cell death (*Figure 1F*). Our findings for B2 RNA SRG upregulation and B2 RNA increased processing coincide with the active neurodegeneration phase of amyloid pathology but in later stages the effect is not that prominent. This suggests that the initial active phase differs from the terminal stage regarding role of B2 RNA. As in the terminal

stage a large number of cells have already died, B2 processing activation appears to be more connected with the initial response of cells to amyloid toxicity.

The B2-RNA-mediated regulation of gene expression during stress identified in one biological context may be relevant to a broad range of cellular types and disease pathophysiology. In the current study, to identify examples of genes that are subject to B2 RNA regulation in hippocampus molecular pathology we utilized the gene list of B2-SRGs identified in our previous study in heat shock. Of the previously identified 1684 B2-SRGs, we found here 72 genes that are deregulated specifically during the active neurodegeneration phase in hippocampal cells. However, this does not exclude the possibility that beyond these 72 genes, B2 RNA processing may also affect expression of additional B2-SRGs that are associated with other hippocampal functions that are independent of amyloid beta pathology. To get a better insight into this possibility, we tested the correlation between the gene expression of other B2-SRGs and the B2 RNA processing ratio, independently of amyloid beta status. In particular, for each of these genes we have calculated the Pearson correlation coefficient between gene expression levels and B2 RNA processing ratio in the hippocampal samples of the current study. As shown in *Figure 8—figure supplement 1E*, we have been able to calculate a statistically significant correlation coefficient for at least 659 genes (the rest where either not sufficiently expressed in all samples or returned a p value > 0.05). From these 659 genes, expression of 344 genes (52.2%) showed a strong correlation with B2 RNA processing ratio ($r \geq 0.5$), while an additional 75 genes (11.3%) showed a weak correlation ($0.5 > r \geq 0.25$), and only 240 genes (36.4%) had an $r < 0.25$ and no correlation. This data suggests that for a large number of the B2-SRGs previously identified in the context of heat shock, correlation between gene expression and B2 RNA processing ratio holds true also in the context of hippocampal cells.

During our amyloid toxicity experiments, cells inoculated with the reverse peptide and the anti-*Hsf1* LNA did not show any reduction in *Hsf1* levels in contrast to those inoculated with amyloid beta (*Figure 8B*). This could be attributed to compensation during non-stress conditions. In contrast, under stress conditions, when *Hsf1* is heavily used due to stress response, cellular needs surpass the available *Hsf1* transcripts that are continuously depleted by the LNA. This is also in agreement with levels of B2-SRGs in this condition (*Figure 8E*), which are minimal in both R-ctrl and R-anti-*Hsf1* conditions and only get activated during stress response to the amyloid beta in the 42-ctrl condition.

Our study leaves a number of open questions. Particularly, our study raises the question whether a similar mode of regulation of SRGs by SINE RNAs may exist also in human and which SINE RNAs could play such a role. Similarly to B2 RNA, such SINE RNAs would be able to bind and inhibit RNA Pol II and would be subject to a similar RNA processing mechanism enabling the release of RNA Pol II. A number of studies have described that in human, repetitive SINE RNAs of the Alu class are also upregulated during cellular stress and can bind RNA Pol II inhibiting the transcription of target genes (*Yakovchuk et al., 2009*). Alu RNAs are widely regarded as the equivalent in human of B2 RNA. Most importantly, as we showed before, human Alu RNAs, alike B2 RNAs, are self-cleaving RNAs and can become destabilized in vitro (*Hernandez et al., 2020*). It remains unknown whether SINE RNAs and *Hsf1* play a similar role in amyloid pathology in the case of humans and whether we can extrapolate the generated conclusions in murine models to deduce that SINE RNAs are key components of the pathophysiological mechanisms underlying debilitating diseases such as AD. One major limitation compared to human pathophysiology is that the phenotype of amyloid pathology is not observed in mice even during aging. Nonetheless, a stress-central role of Alu RNAs, the human counterpart of B2 RNAs is plausible and, thus, future studies need to elucidate whether Alu RNA processing is also hyperactivated in the brain of patients with amyloid pathology in the context of AD.

Moreover, in the RNA-seq data one cannot distinguish between Pol III transcribed B2 RNA and Pol II transcribed B2 RNA (typically embedded within introns and UTRs of mRNAs). To get an indication whether such transcripts may contribute to our data, we have separated the B2 elements against which we map the RNA fragments into two categories, those that fall within exonic/genic regions and those outside these regions (*Figure 3—figure supplement 2*). Although B2 RNAs are produced by multiple copies in the genome, each copy does harbor multiple SNPs, insertions and deletions, which means that each B2 RNA fragment is mapped to a specific set of B2 elements and not to all of them. Thus, despite multiple mapping of the reads, a level of spatial specificity is maintained. If the B2 RNAs we map were coming exclusively from either only Pol III B2 elements or mRNA-embedded B2 elements, we would expect at least some difference in the distribution of

fragments between B2 elements of these two categories, as the second one overlaps with mRNAs. However, as shown in *Figure 3—figure supplement 2*, the fact that distribution models are very similar between the two categories supports the hypothesis that both types of B2 elements may contribute to B2 RNA processing. Thus, it cannot be excluded whether the regulatory role of B2 RNAs may extend from Pol III transcribed B2 RNAs into B2 RNAs embedded into mRNAs (likely nascent ones) that may be also under the same endogenous ribozyme activity of this sequence, may suppress Pol II and get processed in response to stimuli.

Our results suggest that B2 RNA regulation is a new process implicated in response to stress in amyloid pathology but it is definitely not the only one. Since B2-SRGs are highly interconnected and interweaved into various pathways, the impact of SRG hyperactivation by B2 RNA processing may extend to pathways that lie downstream of SRGs and affect various gene programs without binding directly B2 RNA. However, here it should be noted that high levels of *Hsf1* are certainly expected to affect transcription also through Hsf1's conventional transcriptional factor function while there are still many SRGs that are not directly regulated by B2 RNA. Thus, B2 RNA processing described here does not constitute the only one but just one of the parameters in the equation of SRG regulation. It remains unclear which is the interplay between Hsf1 traditional transcription factor activities and its ability to affect B2 RNA processing and whether there is any overlap or synergies. For example, it remains unclear what percentage of activated genes are activated though Hsf1 DNA binding and what through binding of B2 RNA and subsequent release of suppressed Pol II activity.

Moreover, given how easily B2 RNA is processed in the presence of certain proteins, Hsf1 may be only one of the factors accelerating B2 processing in mouse hippocampus as we are just beginning to understand the implications of this form of SINE RNA regulation in cells. A broader role of SINE RNA processing in brain physiology and pathophysiology constitutes, thus, a significant possibility that could further revise our understanding of these RNAs as something more than just transcriptional noise and 'junk DNA' products.

# Materials and methods

## Key resources table

| Reagent type (species) or resource | Designation | Source or reference | Identifiers | Additional information |
|---|---|---|---|---|
| Cell line (*Mus musculus*) | HT-22 | Millipore Sigma | Cat#SCC129, RRID:CVCL_0321 | |
| Sequence-based reagent | Amyloid Beta peptides ( 1-42) | Sigma-Aldrich | Custom synthesis | DAEFRHDSGYEVHHQKLVFFA EDVGSNKGAIIGLMVGGWIA |
| Sequence-based reagent | Amyloid Beta peptides (Reverse 42–1) | Sigma-Aldrich | Custom synthesis | AIVVGGVMLGIIAGKNSGVDEAF FVLKQHHVEYGSDHRFEAD |
| Peptide, recombinant protein | *Hsf1* protein | Enzo life sciences | ADI-SPP-902-F | Synthesized in insect, human sequence |
| Commercial assay, kit | NEBNext Small RNA Library Prep set | NEB | Cat# E7330 | |
| Commercial assay, kit | NEBNext Ultra II directional RNA library prep kit | NEB | Cat# E7760 | |
| Commercial assay, kit | Superscript III RT | Invitrogen | Cat# 18080093 | |
| Commercial assay, kit | Luna universal master mix | NEB | Cat# M3003 | |
| Antibody | *Anti-Hsf1* (Rabbit, polyclonal) | Enzo | Cat# ADI-SPA-901 RRID:AB_10616511 | WB: 1:1000 |

### Amyloid beta peptide preparations

The amyloid beta 1–42 peptides and the respective control peptides (having the reverse aa sequence compared to 1–42 peptides) were synthesized by Sigma Aldrich's custom synthesis service

using the following sequences: DAEFRHDSGYEVHHQKLVFFAEDVGSNKGAIIGLMVGGVVIA (peptide 1-42) and AIVVGGVMLGIIAGKNSGVDEAFFVLKQHHVEYGSDHRFEAD (Reverse). Upon receiving, the peptides were dissolved in 10% $NH_4OH$ at a concentration of 2.1 mg/mL, sonicated for 5 min, aliquoted, dried, and stored at −80˚C for further use (see below).

## Animals and behavioral measurement

Immunohistochemistry was done as described in our previous study (*Mehla et al., 2019*) for the same mice cohorts for which behavioral studies were performed in that study (*Mehla et al., 2019*). In brief, mouse pairs of APP-KI mice carrying Arctic, Swedish, and Beyreuther/Iberian mutations (APPNL-G-F/NL-G-F) were gifted by RIKEN Center for Brain Science, Japan, and the colony of these mice was maintained at Canadian Center for Behavioral Neuroscience vivarium. C57BL/6J mice were used as a WT control and all animals were housed in groups of four mice in each cage in a controlled environment (22˚C–25˚C, 50% humidity and a 12 hr light:dark cycle). All experimental procedures were approved by the institutional animal care committee and performed in accordance with the standards set out by the Canadian Council for Animal Care. In total, we extracted RNA from hippocampi of three different mice per age group per condition. During the hippocampal RNA extractions, the RNA of one of the three 3-month-old wild-type mice had very low RIN scores, which could be a confounding factor for the short-RNA-seq. As this happened some months after the hippocampal extractions, we did not have available other 3-month mice of the same cohort used in our previous study of these mice for the behavioral and IHC studies. Thus, we decided to include only two replicates in this condition. Since the results presented in the current study involve mainly 6-month-old mice we expect the impact to be minimal.

## Cell culture and transfections

HT22 cells, from an immortalized mouse hippocampal cell line (*Davis and Maher, 1994*) (Millipore Cat# SCC129, RRID:CVCL_0321). Cells were provided by the vendor tested negative for infectious diseases by a Mouse Essential CLEAR panel by Charles River Animal Diagnostic Services and verified to be of mouse origin and negative for inter-species contamination from rat, chinese hamster, Golden Syrian hamster, human, and non-human primate (NHP) (as assessed by a Contamination Clear panel by Charles River Animal Diagnostic Services) as well as negative for mycoplasma contamination. Cells were cultured in DMEM (Sigma) and 1% Penicillin/Streptomycin (Gibco). Cells were thawed and passaged at least 3 days before the transfection date in order to allow sufficient time for cells to recover from the stress of cryopreservation and not interfere with the assessment of cellular response to stress in subsequent experiments. For knocking down of *Hsf1* mRNA levels, we used an LNA long RNA GapmeR against *Hsf1* (Exiqon/Qiagen) with the following sequence: 5′-′GAAGGA TGGAGTCAA-3′ and an LNA long RNA GapmeR with the following scramble sequence: 5′-CCTCAA TTTTATCAC-3′. For knocking down B2 RNA transcripts, we used an anti-B2 LNA pool consisting of five LNAs, synthesized using the K and E DNA and RNA synthesizer: 1. 5′ - G*T*T*A*CGGATGG TT*G*T*G* - 3′; 2. 5′- A*G*A*T*CTCATTACA*G*A*T*- 3′; 3. 5′ - A*G*A*T*CTCATTACG*G*A*T* - 3′′; 4. 5′ - A*G*A*T*CCCATTACA*G*A*T* - 3′; 5. 5′ - A*G*A*T*CCCATTACG*G*A*T* - 3′; 6. 5′-T*G *TAGCTGTCTTCA*G*−3′. Nucleotides were ordered from Sigma Aldrich as DMT phosphoramidites. The day of LNA transfections, following 5-min incubation with TrypLE Express Enzyme (Gibco) (1x), cells were passaged and transferred to a six-well plate at a 100,000 cells/well density and LNA transfections were performed simultaneously, using the HiPerfect reagent (Qiagen). Transfection was performed as follows: Firstly, LNAs were reconstituted in nuclease-free water to 50 µM. Subsequently, 3 µL 50 µM LNA were mixed and incubated with 4 µL Hiperfect reagent and 30 µL nuclease-free water, at room temperature for 20 min, and then added drop-wise to cells that had just been plated in 1 ml of medium/well (still not attached) to a final LNA concentration of 150 nM. For anti-*Hsf1* LNAs transfections, incubation with amyloid beta (peptide 1-42) and control peptides (Reverse) was performed 24 hr after transfection with LNAs. Peptides were initially dissolved in DMSO for incubation at 37˚C for 1 hr and then added to cells to a final concentration of 30 µM for 6 hr before treating with 0.5 mL TrypLE (Gibco), pelleting cells at 1000 rpm for 5 min and resuspending the pellet in 1 mL Trizol reagent (Thermofisher) for RNA extraction based on Manufacturer's instructions. For anti-B2 LNA, transfection and incubation lasted 6 hr before RNA extraction.

## Reverse transcription and quantitative PCR

RNA was extracted using Trizol reagent as described and reverse transcribed using Superscript III (18080093, Invitrogen) by the following method: 50 ng total RNA was mixed with 100 ng random primers (C1181, Promega) and 1 µL of 10 mM dNTP mix. The mixture was incubated for 5 min at 65° C and placed immediately on ice. The mixture was then incubated with 4 µL of 5x First strand buffer, 1 µL of 0.1M DTT and 0.4 µL Superscript III (18080093, Invitrogen) for 5 min at 25°C, 60 min at 55°C and 15 min at 70°C. cDNA was analyzed by qPCR using 2 µL of 1:20 diluted cDNA, 0.5 µL of 10 µM of each gene-specific primer, 2 µL H$_2$O and 5 µL of Luna Universal qPCR Master Mix (M3003E, NEB). Thermocycler conditions are as follows: 3 min at 95°C (15 s at 95°C, 30 s at 54°C, 30 s at 66°C) × 40 cycles. Fluorometer readings were taken during extension and qPCR was performed using the Bio-Rad CFX384 Real-time detection system. Standard curves were prepared for relative expression and the analysis of PCR efficiency by pooling 2 µL of each cDNA sample and standard diluting SD1: 1:5, SD2: 1:10, SD3: 1:20, SD4: 1:40. Samples were either analyzed by standard curve relative expression or $2^{-CT}$ fold change analysis. Student's T-test were used to study significance as described in the respective figure legends. Primers were ordered from IDT as custom oligos and are listed in *Appendix 1—table 1*.

## RNA in vitro transcription and RNA-protein incubations

B2 template for in vitro RNA transcription was ordered as IDT g-block (lower case denotes the T7 promoter sequence): 5'- taatacgactcactata GGGGCTGGTGAGATGGCTCAGTGGGTAAGAG-CACCCGACTGCTCTTCCGAAGGTCCGGAGTTCAAATCCCAGCAACCACATGGTGGCTCA-CAACCATCCGTAACGAGATCTGACTCCCTCTTCTGGAGTGTCTGAAGACAGCTACAGTGTAC TTACATATAATAAATAAATAAATCTTTAAAAAAAAA - 3'. The template was amplified by PCR using a T7 promoter sequence as the forward primer: 5'-TAATACGACTCACTATAG and the following sequence as reverse primer: 5'-TTTTTTTTTAAAGATTTATTTATTTATTATATGTAAGTACA. B2mut4b was transcribed from the following template (same primers): taatacgactcactataGGGCTGGTGAGA TGGCTCAGTGGGTAAGAGCACCCGACTGCTCTTCCGAAGGTCCGGAGTTCAAATCCCAGCAAC-CACATGGTGGCTCACAACCATCCGTAACGAGATCTGACTCCCTCTTCTTCTGAAGACAGCTACAG TGTACTTACATATAATAAATAAATAAATCTTTAAAAAAAAA. Primers were diluted to 10 mM and PCR was performed using the NEB Q5 polymerase, Q5 reaction buffer (10x), Q5 high GC enhancer (10x). The reaction proceeded at hot start 98°C – 30 s (98°C – 5 s, 58°C – 10 s, 72°C – 10 s) X35 cycles, 72°C – 10 min. The samples were then analyzed by agarose gel electrophoresis (Bio-Rad, 1613100EDU) and the bands were gel extracted at the desired size and purified using the BioBasic EZ-10 gel extraction kit (BS353). A subsequent PCR was then repeated and 1 µg of the amplified g-block was then in vitro transcribed by T7 RNA polymerase (NEB, M0251) for 2 hr at 37°C. The reaction was buffered using the T7 RNA polymerase buffer in addition to 10 mM NTPs (ATP: P1132, CTP: P1142, GTP: P1152, UTP: P1162) in a final 20 µL reaction. RNA was purified using the Zymo Research RNA Clean and Concentrator - 25 kit. Sequences of RNA controls used were as follows: Control #1(G-44U): 5'GCCCCGUUGCAAUGGAAUGACAGCGGGUAUGUUAAACAACCCCAUCCG UCAUGGAGACAGGUGGACGUUAAAUAUAAACCUGAAGAUUAAACAUGACUGAAUCUUUUGC UACUAGAAUGGUGAGCAAGGGCGAGGAGCUGUUC 3', control #2:(5' Zika UTR scramble):5' UACAAUCACGAAAGUCAAUUAUAGUUUCGAGUCGUAACGAGAACAUUUCCCGCGGACCAA UUUAAGGAGUAACUAAAGUGUGAAAUGAUUCCGGAAUACUGUUGAAAUUGCGGAUCGAGC UUGCAGCCGUUAAAUUACCGGACGUUAGUGAAGUGCAGAUAUG 3'.

B2 RNA re-folding and B2 RNA – Hsf1 incubations were performed as described previously (14). In brief, in vitro-transcribed B2 RNA was folded with 300 mM NaCl through incubation for 1 min at 50°C and cooling at a rate of 1°C/10 s until 4°C. Subsequently, B2 RNAs were incubated at a final concentration of 0.4 µM unless otherwise stated with the addition of Hsf1 diluted in TAP buffer (final reaction concentrations: 5 nM Tris pH 7.9, 0.5 mM MgCl$_2$, 0.02 mM EDTA, 0.01% NP-40, 1% glycerol, 0.2 mM DTT). Hsf1 protein incubations were performed with phosphorylated, recombinant, His-tagged Hsf1 (Enzo Life Sciences: ADI-SPP-902-F; ~60KDa). Hsf1 working concentrations were 250 nM unless otherwise specified, diluted in TAP buffer. PNK (NEB, M0201; ~35KDa) was used as a control protein because it is an RNA-binding protein that is used in the construction of our short RNA libraries and we wanted show that short RNA seq data are free of such confounding factors that could potentially generate artificial fragments. PolyA (NEB, M0276; ~56KDa) was used as

control protein because it is an RNA-binding protein that should not process B2 RNAs. Denatured Hsf1 was prepared by diluting in TAP buffer as above and heating at 95°C for 3 min; denatured Hsf1 was used as a control to ensure buffer components of Hsf1 themselves were not responsible for B2 RNA processing.

Fragmentation of B2 RNA was analyzed on 8M urea 10% PAGE gels stained by SYBR II (Invitrogen, S7564). Gel analysis occurred on Amersham Typhoon instruments. Band absorbance was analyzed using ImageJ area under the curve software and normalized by the ratio of experimental over initial as described previously (*Zovoilis et al., 2016*).

## Short-RNA-seq and long-RNA-seq

Using the miRvana miRNA size selection kit (Thermo Fisher) as described before, 1.5 µg total RNA was size separated into short and long fractions (*Zovoilis et al., 2016*). In brief, following addition of the lysis/binding buffer and the homogenate additive solution to the RNA, 1/3 of the volume 100% EtOH was added and the mix was passed through the column for binding long RNAs. 100% EtOH at 2/3 of the flow through volume was subsequently added to the flow through and passed through a second column for binding short RNAs. Eluted RNAs were tested for size and quality using the Agilent Bioanalyzer RNA pico-kit. For long-RNA-seq, the long RNA fractions were cleaned and concentrated using the RNeasy Minelute kit (Qiagen) and ribodepleted using the rRNA depletion kit (NEB). The library was then prepared using the NEBNext Ultra II direction Library preparation kit (NEB, E7760), and sample cleanups were performed using the Omega NGS Total Pure Mag Beads (Omega, SKU: M1378-01) 0.5X and 1.2X before library amplification and 0.9X following amplification. nine cycles were used during amplification. For short-RNA-seq, the short RNA fractions were subjected to 3′-phosphoryl removal for 1 hr at 37°C, treated with the T4 PNK enzyme (NEB, M0201), using exclusively the 10X PNK buffer (NEB). The short fractions were cleaned and concentrated using the RNeasy Minelute kit and the library was prepared using the NEBNext small RNA library prep set (E7330) as described before (*Zovoilis et al., 2016*). Sample cleanups were performed using again the Omega NGS Total Pure Mag Beads (Omega) 1.2X following library amplification. Library amplification used 15 cycles. Quantification of libraries was done by qPCR using the NEBNext library quant kit for Illumina (NEB, E7630) and library sizes were analyzed using the Agilent bioanalyzer 2100 HS DNA kit. Equimolar amounts were prepared for sequencing. Libraries were sequenced on an Illumina HiSeq platform using 150nt read lengths. For B2 RNA in vitro sequencing, a 10X reaction as described of B2 RNA in solution with Hsf1 was incubated for 60 min at 37°C. The RNA was prepared as described above at twice the concentration of the recommended kit components. RNA was cleaned and concentrated using the modified RNeasy minElute kit.

## Western blotting

HT22 cells were seeded at 100,000 cells per well and transfected with LNA in a six-well plate as described. Cells were harvested and lysed using RIPA lysis buffer with protease and phosphatase inhibitors added. Next, the cell suspension was centrifuged at 13,000 rpm for 20 min at 4°C, supernatant was aspirated and pellet discarded. Protein concentration was determined using a Bradford assay (bio-rad; 500–0205). Equal amounts of soluble protein were loaded (25 µg) for resolving with SDS-PAGE. Proteins were transferred onto nitrocellulose membrane (GE healthcare), blocked for 1 hr with 1% milk in 0.02% PBST at room temperature. Individual proteins were detected using polyclonal rabbit HSF1 antibody (1:1000; Enzo Life Sciences; ADI-SPA-901; RRID:AB_10616511). After incubating primary antibody, blots were washed using 0.1% PBST and reprobed with anti-rabbit-HRP conjugate secondary antibody (1:5000; abcam; ab97051). Proteins were visualized using Pierce enhanced chemiluminescence detection system (Thermo Fisher Scientific; 32106). Blots were imaged in an AI600 imager (GE Healthcare) and densitometry performed using ImageJ.

## Bioinformatics analysis

For the tissue enrichment and GO term analysis of B2-SRGs (*Supplementary file 1*), we used the DAVID function annotation platform (DAVID 6.8, February 2020) with default parameters (EASE score 0.1, max. 1000 entries), and a reporting EASE score threshold of 0.05 and p-adjusted values calculated based on the Benjamini method (*Huang et al., 2009a*; *Huang et al., 2009b*). For both GO Biological process and Cellular Compartment, we selected the BP- or CC-direct options.

For the analysis of the short-RNA-seq and long-RNA-seq data, initially FastQC (Babraham Bioinformatics, https://www.bioinformatics.babraham.ac.uk/projects/fastqc/) was run for quality control of generated reads in fastq format. Subsequently, standard Illumina adaptor sequences were trimmed off using cutadapt-1.18 (https://doi.org/10.14806/ej.17.1.200). Short-RNA-seq reads were mapped to mouse reference genome (UCSC mm10) (November 2017) using bwa-0.7.17 in single-end mode with default aln parameters (*Li and Durbin, 2009*). Long-RNA-seq reads for each sample were mapped to reference genome ensembl GRCm38 (November 2018) primary assembly using hisat2-2.1.0, in single-end mode, with the following parameters: Report alignments tailored for transcript assemblers including StringTie, searches for at most one distinct, primary alignments for each read (*Kim et al., 2019*). SAM format files generated from mapping were converted to BAM format files using samtools-1.6 (*Li et al., 2009*), and to files in BED format with bamToBed utility from BEDTools-2.26.0 (*Quinlan and Hall, 2010*).

Models of distribution of 5′ end read fragments within the B2 loci (B2_Mm1a, B2_Mm1t and B2_Mm2) were performed using an in house python script. In brief, the script constructs a read accumulation metagene model around a hypothetical set of genomic points, in our case the start site for all B2 elements (TSS), in which the numbers of reads (or read 5′ ends) around each different TSS were calculated and attributed to defined points in the model. B2 element coordinates are based on the UCSC genome browser RepeatMasker track (as of November 2018). To calculate the B2 processing ratio, Babraham NGS analysis suite Seqmonk 1.38.2 (https://www.bioinformatics.babraham.ac.uk/projects/seqmonk/) was used to obtain number of long reads overlapping with B2 loci (B2_Mm1a, B2_Mm1t and B2_Mm2), as well as the number of reads overlapping with tRNA loci from −5 to 15 bp. Processing ratio for each sample was calculated by processed B2 count obtained from the in house python scripts normalized by tRNA from −5 to 15 bp and small reads fastq read count, as well as B2 count and long reads fastq read count: [Small fragments (position 95–110)/[tRNAs/small RNA fastq]]/[B2RNA/long RNA fastq]. The full length of B2 consensus sequence is 188nt, and this is the one we use for the in vitro experiments. However, structure of the RNA has been resolved only for the 155nt (*Espinoza et al., 2007*), and this is the structure currently used in our figures. For the mapping of short fragments, we have used the same range tested in our previous study (*Zovoilis et al., 2016*) to maintain consistency of the results. The reason why this 120nt threshold was selected in the Cell paper was to exclude artifacts from short RNAs mapping partially in our metagene as well as downstream of those B2 elements that are shorter from the consensus sequence.

In long-RNA-seq, FPKM (Fragments Per Kilobase of transcript per Million) and TPM (Transcripts Per Million) for genes were generated using StringTie-1.3.4d (*Pertea et al., 2015*) with the following annotation: ensembl GRCm38 patch 94 gff3 file, and parameters limiting the processing of read alignments to only estimate and output the assembled transcripts matching the reference transcripts given in annotation and excluding non-regular chromosomes. Because TPM already includes scaling of the data it is unsuitable for the averaging of the gene expression levels of multiple genes (B2-SRGs) used in the boxplots of *Figure 2*. This does not apply in case of single genes as in *Figure 2C* (*Trp53*) or in the heatmap of the same figure, where each gene is presented in a separate row, and for which TPM values are used. For data visualization, statistics and differential expression analysis, we employed R (version 3.4.3) (https://www.R-project.org/) and the package DESeq2 (*Love et al., 2014*). Differential expression analysis was implemented on transcript count data for 6-month-old mice between APP and wild type. Boxplots central line represents median and t-test was applied on the group numbers mentioned in the text. PCA plots for samples used and read count correlation matrix between 6-month-old mice samples are presented in *Figure 6—figure supplement 1*.

For *Hsf1* metagene analysis, we used peak.txt files of Hsf1 peaks for ChIP-seq from *Mahat et al., 2016*. Peaks were analyzed with Seqmonk around Transcription start sites of genes (TSS) based on the Eponine annotation (*Down and Hubbard, 2002*), and filtered based on their overlap with B2 RNA regulated genes (*Zovoilis et al., 2016*). Also, B2 RNA-binding (CHART-seq) peaks were analyzed with Seqmonk around TSS (Eponine), then filtered by overlapping with learning-associated SRGs or all genes (*Peleg et al., 2010*). Relative density metagene plots of the distribution of the above peaks were generated using Seqmonk.

## Data access

Short- and long-RNA-seq raw data have been deposited to GEO with access number GSE149243.

## Acknowledgements

This work has been supported by an Explorations Grant # 201700011 to AZ and MM from Alberta Innovates and the Alberta Prion Research Institute, a Grant # 201900003 to AZ and MM from the Alzheimer Society of Alberta and Northwest Territories and the Alberta Prion Research Institute, a Discovery Grant # RGPIN-2018–05955 to AZ from NSERC, the BioNet Alberta grant to AZ from Genome Canada and a Compute Canada Resource Allocation Grant to AZ. AZ is supported by the Canada Research Chairs Program and the Canada Foundation for Innovation and is a former EMBO and DFG long-term fellow. YC is supported by an Alberta Innovates (AITF) fellowship. LS is supported by the AMR One Health grant by the Government of Alberta. We are grateful to Dr. Angeliki Pantazi for extensively reviewing, editing and commenting on the manuscript. We are grateful to the Patel Lab (Trushar Patel, Darren Gemmill) and Wieden Lab (Justin Vigar, Hans Joachim Wieden) at the University of Lethbridge for providing us with the RNA controls #1 and #2.

## Additional information

### Funding

| Funder | Grant reference number | Author |
|---|---|---|
| Alberta Innovates | 201700011 | Majid H Mohajerani<br>Athanasios Zovoilis |
| Alberta Prion Research Institute | 201700011 | Majid H Mohajerani<br>Athanasios Zovoilis |
| Alzheimer Society of Alberta and Northwest Territories | 201900003 | Majid H Mohajerani<br>Athanasios Zovoilis |
| Alberta Prion Research Institute | 201900003 | Majid H Mohajerani<br>Athanasios Zovoilis |
| Natural Sciences and Engineering Research Council of Canada | RGPIN-2018–05955 | Athanasios Zovoilis |
| Genome Canada | BioNet Alberta grant | Athanasios Zovoilis |
| Compute Canada | Resource Allocation Grant | Athanasios Zovoilis |
| Canada Research Chairs | | Athanasios Zovoilis |
| Canada Foundation for Innovation | | Athanasios Zovoilis |
| Alberta Innovates | (AITF) fellowship | Yubo Cheng |
| Government of Alberta | AMR One Health grant | Luke Saville |

The funders had no role in study design, data collection and interpretation, or the decision to submit the work for publication.

### Author contributions

Yubo Cheng, Data curation, Software, Formal analysis, Validation, Investigation, Visualization, Methodology, Writing - original draft, Writing - review and editing; Luke Saville, Data curation, Formal analysis, Validation, Investigation, Visualization, Methodology, Writing - original draft, Writing - review and editing; Babita Gollen, Investigation, Methodology, Project administration; Christopher Isaac, Data curation, Formal analysis, Investigation, Methodology; Abel Belay, Investigation, qPCR assays; Jogender Mehla, Resources, Methodology; Kush Patel, WB assays; Nehal Thakor, Resources, WB assays; Majid H Mohajerani, Resources, Supervision, Project administration; Athanasios Zovoilis, Conceptualization, Resources, Data curation, Software, Formal analysis, Supervision, Funding acquisition, Validation, Investigation, Visualization, Methodology, Writing - original draft, Project administration, Writing - review and editing

## Author ORCIDs
Luke Saville (ID) https://orcid.org/0000-0002-9480-1268
Majid H Mohajerani (ID) https://orcid.org/0000-0003-0964-2977
Athanasios Zovoilis (ID) https://orcid.org/0000-0001-6425-0402

## Ethics
Animal experimentation: All experimental procedures were approved by the institutional animal care committee protocol number1404 and performed in accordance with the standards set out by the Canadian Council for Animal Care.

## Decision letter and Author response
Decision letter https://doi.org/10.7554/eLife.61265.sa1
Author response https://doi.org/10.7554/eLife.61265.sa2

# Additional files

### Supplementary files
• Supplementary file 1. Supplementary tables. Supplementary Table 1. List of B2 RNA regulated SRGs(B2-SRGs). Data are compiled from *Zovoilis et al., 2016* and include those genes that are close to B2 CHART peaks (genome-binding sites) before but not after the application of stress stimulus. Supplementary Table 2. Complete lists of enriched terms in B2 RNA regulated SRGs(B2-SRGs)(see Suppl.Table 1) for Tissue Enrichemnt (left), Biological Process (middle) and Cellular Compartment (right). Supplementary Table 3. List of B2 RNA regulated SRGs (B2-SRGs) (see Suppl.Table 1) that are associated with learning based on *Peleg et al., 2010*. Supplementary Table 4. Upregulated genes in hippocampi of APP 6-month-old mice compared to 6-month WT mice. Values were calculated using DESeq (see Materials and methods) on long-RNA-seq data. Only genes with an FDR < 0.2 are depicted. Supplementary Table 5. List of B2 RNA regulated SRGs (B2-SRGs) (see Suppl.Table 1) that are upregulated in 6-month-old APP mice compared to WT (see Suppl.Table 4) Supplementary Table 6. List of B2 RNA regulated SRGs (B2-SRGs) (see Suppl.Table 1) that are upregulated in 6-month-old APP mice (see Suppl.Table 4) and are associated with learning based on *Peleg et al., 2010*. Supplementary Table 7. Complete lists of enriched terms in B2 RNA regulated SRGs (B2-SRGs) that are upregulated in 6-month-old APP mice compared to WT (see Suppl.Table 5) for Biological Process (left) and Cellular Compartment (right). Supplementary Table 8. Upregulated genes in HT22 cells treated with amyloid beta and Scr LNA compared to cells treated with the control peptide and scr LNA. Values were calculated using DESeq (see Materials and methods) on long-RNA-seq data. Only genes with an FDR < 0.2 are depicted. Supplementary Table 9. List of genes that are upregulated in HT22 cells treated with amyloid beta (see Suppl.Table 8) and in 6-month-old APP mice (see Suppl.Table 4) Supplementary Table 10. List of B2 RNA regulated SRGs (B2-SRGs) (see Suppl.Table 1) that are upregulated in HT22 cells treated with amyloid beta and Scr LNA compared with cells treated with the control peptide and scr LNA (see Suppl.Table 8 ) Supplementary Table 11. Complete lists of enriched terms in B2 RNA regulated SRGs (B2-SRGs) that are upregulated in HT22 cells treated with amyloid beta (see Suppl.Table 10) for Biological Process (left) and Cellular Compartment (right). Supplementary Table 12. Correlation co-efficients and p-values for genes of *Figure 8—figure supplement 2*. Includes genes for which there was readcoverage across all sample and the correlation p value was less than 0.05. Supplementary Table 13. List of non-B2 RNA regulated genes (random set) used throughout the study.

• Transparent reporting form

### Data availability
Short and long-RNA-seq data have been deposited to GEO with access number GSE149243.

The following dataset was generated:

| Author(s) | Year | Dataset title | Dataset URL | Database and Identifier |
|---|---|---|---|---|
| Cheng Y, Saville L, Zovoilis A | 2020 | Increased processing of SINE B2 non coding RNAs unveils a novel type of transcriptome de-regulation underlying amyloid beta neuro-pathology | https://www.ncbi.nlm.nih.gov/geo/query/acc.cgi?acc=GSE149243 | NCBI Gene Expression Omnibus, GSE149243 |

The following previously published dataset was used:

| Author(s) | Year | Dataset title | Dataset URL | Database and Identifier |
|---|---|---|---|---|
| Mahat DB, Sala-manca HH, Duarte FM, Danko CG, Lis JT | 2016 | Mammalian Heat Shock Response and Mechanisms Underlying Its Genome-wide Transcriptional Regulation | https://www.ncbi.nlm.nih.gov/geo/query/acc.cgi?acc=GSE71708 | NCBI Gene Expression Omnibus, GSE71708 |

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

# Appendix 1

**Appendix 1—table 1.** DNA/RNA sequences.

| Reagent type (species) or resource | Designation | Source or reference | Identifiers | Additional information |
|---|---|---|---|---|
| Sequence-based reagent | *FosB* Forward | IDT | Custom | 5-CGAGCTGCAAAAAGAGAAGG −3 |
| Sequence-based reagent | *FosB* Reverse | IDT | Custom | 5- TTACAGAGCAAGAAGGGAGG −3 |
| Sequence-based reagent | *Pag1* Forward | IDT | Custom | 5-GAGCACAACTTCAAAGCTGG-3 |
| Sequence-based reagent | *Pag1* Reverse | IDT | Custom | 5- TCATCAGGTTCTCATGGTCC −3 |
| Sequence-based reagent | *Sema5a* Forward | IDT | Custom | 5- ATGAGGCTGTGCAGTTCAGT-3 |
| Sequence-based reagent | *Sema5a* Reverse | IDT | Custom | 5-GTAACCAGGGGCCAATTTCT-3 |
| Sequence-based reagent | *Sgms1* Forward | IDT | Custom | 5- ACCATAGACCACACAGGCTA-3 |
| Sequence-based reagent | *Sgms1* Reverse | IDT | Custom | 5- TTTCTTCCGGTCTGAGCACT-3 |
| Sequence-based reagent | *Hsf1* Forward | IDT | Custom | 5- TGACACCGAGTTCCAGCATC-3 |
| Sequence-based reagent | *Hsf1* Reverse | IDT | Custom | 5- TGACACTGTCCTGGCGTATT-3 |
| Sequence-based reagent | *Mitf* Forward | IDT | Custom | 5- AAGCTCAGAGGCACCAGGTA-3 |
| Sequence-based reagent | *Mitf* Reverse | IDT | Custom | 5- CCTGCTCTGCTCCTCAAACT-3 |
| Sequence-based reagent | *7SK* Forward | IDT | Custom | 5-GACATCTGTCACCCCATTGA-3 |
| Sequence-based reagent | *7SK* Reverse | IDT | Custom | 5- GCCTCATTTGGATGTGTCTG-3 |
| Sequence-based reagent | *Hprt* Forward | IDT | Custom | 5- TCCTCCTCAGACCGCTTTT-3 |
| Sequence-based reagent | *Hprt* Reverse | IDT | Custom | 5- CCTGGTTCATCATCGCTAATC-3 |

*Continued on next page*

Appendix 1—table 1 continued

| Reagent type (species) or resource | Designation | Source or reference | Identifiers | Additional information |
|---|---|---|---|---|
| Sequence-based reagent | B2 Forward | IDT | Custom | 5- GGGGCTGGTGAGATG-3 |
| Sequence-based reagent | B2 Reverse | IDT | Custom | 5-AGCTGTCTTCAGACACTCC −3 |
| Sequence-based reagent | Adcy1 Forward | IDT | Custom | 5- GCATGACAATGTGAGCATCC −3 |
| Sequence-based reagent | Adcy1 Reverse | IDT | Custom | 5-TCAAGTCCCATCTCCACACA −3 |
| Sequence-based reagent | Kcnq3 Forward | IDT | Custom | 5- AGCACCGTCAGAAGCACTTT −3 |
| Sequence-based reagent | Kcnq3 Reverse | IDT | Custom | 5-TCCAAGAGACCCAGCTTTTG-3 |
| Sequence-based reagent | Klf15 Forward | IDT | Custom | 5-TCATGGAGGAGAGCCTCTGT-3 |
| Sequence-based reagent | Klf15 Reverse | IDT | Custom | 5-TCCAAGAGACCCAGCTTTTG-3 |
| Sequence-based reagent | Magi2 Forward | IDT | Custom | 5-CGGGATCACACTTTTCACCT-3 |
| Sequence-based reagent | Magi2 Reverse | IDT | Custom | 5-CGGGATCACACTTTTCACCT-3 |
| Sequence-based reagent | Palld Forward | IDT | Custom | 5-CAGTGGCTCAGACAGCACAT-3 |
| Sequence-based reagent | Palld Reverse | IDT | Custom | 5-CTCCTGTTTTCGGAGCTGAG-3 |
| Sequence-based reagent | Enpp2 Forward | IDT | Custom | 5-GACTGTCGGTGTGACAACCT-3 |
| Sequence-based reagent | Enpp2 Reverse | IDT | Custom | 5-CTTCTGAGCAGTGACAGGCA-3 |
| Sequence-based reagent | RPS15 Forward | IDT | Custom | 5-AACCAGAGATGATCGGCCAC-3 |
| Sequence-based reagent | RPS15 Reverse | IDT | Custom | 5-ATGAATCGGGAGGAGTGGGT-3 |
| Sequence-based reagent | Calm2 Forward | IDT | Custom | 5-GACTGAAGAGCAGATTGCAG-3 |
| Sequence-based reagent | Calm2 Reverse | IDT | Custom | 5-CAGTTCTGCTTCTGTGGGGT-3 |

*Appendix 1—table 1 continued*

| Reagent type (species) or resource | Designation | Source or reference | Identifiers | Additional information |
|---|---|---|---|---|
| Sequence-based reagent | *Kalrn* Forward | IDT | Custom | 5-CCCTGAACTCCATCCACAGT-3 |
| Sequence-based reagent | *Kalrn* Reverse | IDT | Custom | 5-GAGGGGTGTGTGTGACTCTT-3 |
| Sequence-based reagent | B2 RNA | IDT G-block | Custom synthesis. *Zovoilis et al., 2016* | 5'- taatacgactcactata GGGGCTGGTGAGATGGCTCAGT GGGTAAGAGCACCCGACTGCTCTTCCGAAGGTCCGGAG TTCAAATCCCAGCAACCACATGGTGGCTCACAACCATCC GTAACGAGATCTGACTCCCTCTTCTGGAGTGTCTGAAGA CAGCTACAGTGTACTTACATATAATAAATAAATAA ATCTTTAAAAAAAAA - 3 |
| Sequence-based reagent | B2mut4b | IDT G-block | Custom synthesis. | taatacgactcactataGGGCTGGTGAGATGGCTCAGTGG GTAAGAGCACCCGACTGCTCTTCCGAAGGTCCGGAG TTCAAATCCCAGCAACCACATGGTGGCTCACAACCAT CCGTAACGAGATCTGACTCCCTCTTCTTCTGAAGACAG CTACAGTGTACTTACATATAATAAATAAA TAAATCTTTAAAAAAAAA |

