## [Decision Letter]

[Editors' note: this paper was reviewed by Review Commons.]

**Acceptance summary:**

The revised manuscript addresses the reviewer's comments and includes new data supporting the discovery that amyloid beta neuropathology involves deregulation of SINE B2 ncRNAs. The burden of Alzheimer's disease continues to increase, and novel insights into the mechanisms by which the pathogenic amyloid peptide exerts its toxic effects on neuronal tissues are appreciated and necessary. This manuscript reveals a previously unreported phenomenon elicited by the amyloid peptide leading to activation of cell death pathways, which supported by results from animal models and in vitro cell cultures.

**Decision letter after peer review:**

Thank you for submitting your article "Increased processing of SINE B2 non coding RNAs unveils a novel type of transcriptome de-regulation underlying amyloid beta neuro-pathology" for consideration by *eLife*. Your article has been reviewed by two peer reviewers at Review Commons, and the evaluation has been overseen by a Reviewing Editor and James Manley as the Senior Editor.

The reviewers have discussed the reviews with one another and the Reviewing Editor has drafted this decision to help you prepare a revised submission.

After thorough discussion, the reviewers agreed that the manuscript by Zovoilis et al. is much improved from the original submission to Review Commons and that the authors have addressed many of the original concerns raised by reviewers. However, two major concerns remain, and reviewers agreed to encourage a resubmission addressing these.

B2 RNAs encoded from SINE B2 elements have been directly implicated in stress response by its inherent ability to bind RNA Pol II and suppress stress response genes (SRG) in homeostatic conditions. However, upon stimuli, B2 RNAs are cleaved and degraded, resulting in the release of RNA polymerase II (RNAPII) and upregulation of SRGs. Previous work from the senior author of this manuscript identified the Polycomb repressive complex 2 (PRC2) component EZH2 to be the B2 RNA processing factor, cleaving B2 and releasing RNAPII. SRGs are upregulated upon stress, for example in age associated neuropathologies like Alzheimer's disease (AD). Considering that hippocampus is a primary target of amyloid pathologies and given that SRGs are suggested to be key for the function of a healthy hippocampus, the authors set out to understand the role of B2 RNAs that are linked to SRG regulation in the mouse hippocampus with amyloid pathology. They use disease relevant in vivo and in vitro models combined with unbiased RNA seq data analysis for this endeavor, which indicate the potential relevance of B2 RNAs in APP-mediated neuronal pathologies in mice while also identifying Hsf1 as the factor cleaving B2 RNAs in the hippocampus. The work is deemed interesting and identification of Hsf1 as the processing factor for B2 RNAs in the hippocampus is significant.

1) Reviewers remain concerned that the bulk of the analyses relies on genes that were identified as B2-regulated SRGs in a prior experimental system (heat-shocked NIH3T3 cells) that is completely different from the amyloid pathology models used here. The authors sought to address this issue in the revised manuscript, but questions remain. Indeed, the new Supplementary tables 4 and 5 in Supplementary file 1 show that of the ~1600 B2-SRGs identified in NIH3T3 cells, only 72 show expression patterns consistent with the regulatory model proposed; moreover, this was using FDR<0.2. How many genes would be left with FDR<0.05? The authors did include important new data in the HT22 cell model showing that mRNA levels for four genes increase after treatment with a B2-targeted LNA. These data are compelling, but a few additional controls are needed. Figure 6F needs negative controls showing qRT-PCR of genes that are not thought to be B2-SRGs. In Figure 6E, it is important that the full length B2 RNA is being detected (i.e. a PCR product ~180 nt) since their model states that these genes are repressed by full length B2 RNA prior to its degradation. The data in Figure 6 support the model that these four genes are under B2 control, but they don't show the relationship with amyloid pathology. What do the expression patterns of the 4 genes in Figure 6F look like in the mouse RNA-seq data (3m, 6m, 12m, WT vs APP)? Does the amyloid beta peptide treatment of the HT22 cells no longer induce expression of these four genes in the presence of the B2 LNA?

2) Reviewers remain concerned about the strength of the data regarding the role of Hsf1 in B2 RNA processing (although it seems like the authors are currently working on important control experiments, which might alleviate reviewer's concerns.) The negative controls with recombinant proteins prepared similarly to Hsf1, and with similarly sized control RNAs are critical. In addition, the full gels for these experiments need to be shown so the formation of short B2 RNA products can be evaluated in conjunction with the loss of the full length B2 RNA. This will help distinguish between specific processing controlled by the B2 ribozyme/Hsf1 activity versus non-specific breakdown. Moreover, the sizes of the in vitro processed products should correlate with the 5' end peaks from Figure 3A if the cellular and in vitro processing indeed arises from the same mechanism.

---

## [Author Response]

Reviewer 1:Reviewer #1 (Evidence, reproducibility and clarity Required):B2 RNAs, encoded from SINE B2 elements has been directly implicated in stress response by its inherent ability to bind RNA Pol II and suppress stress response genes (SRG) in homeostatic conditions. However, upon stimuli, B2 RNAs are cleaved and degraded, resulting in the release of RNA pol II and upregulation of SRGs. Previous work from the senior author identified PRC2 component EZH2 to be the B2 RNA processing factor, cleaving B2, and releasing POL2. SRGs are upregulated upon stress, for example in age-associated neuropathologies like Alzheimer's disease (AD). Considering that the hippocampus is a primary target of amyloid pathologies as well as since SRGs are suggested to be key for the function of a healthy hippocampus, the authors set to understand the role of B2 RNAs that are linked to SRG regulation in the mouse hippocampus with amyloid pathology. They use disease-relevant in vivo and in vitro models combined with unbiased RNA seq data analysis for this endeavor, which indicates the potential relevance of B2 RNAs in APP mediated neuronal pathologies in mice as well as identifies Hsf1 as the factor cleaving B2 RNAs in the hippocampus.

This reviewer generally remarks that “*The work is interesting and identification of Hsf1 as the processing factor for B2 RNAs in the hippocampus is significant. I would like to credit the authors for their elegant* in vivo *experimental design in Figure 2.”*

We appreciate the encouraging comments made by this reviewer.

General comment: The reviewer finds “some of the conclusions to be overstated” and has brought a number of concerns to our attention. Indeed, we agree that provision of additional data and details is needed to avoid any confusion about the gene pathways to which our findings apply. In the initial manuscript, (Figures 2 D, F and 6 D, F), we presented the gene expression levels of all B2 RNA regulated SRGs identified in our previous study (Zovoilis et al., 2016), referred as B2 RNA regulated SRGs or B2-SRGs throughout the manuscript. To this end, we performed the respective statistical tests between the different conditions considering these genes, in order to show the transcription dynamics of these genes in either amyloid beta pathology (APP mice /Figures 2D, F) or amyloid beta toxicity (HT22 cells / Figures 6D, F). Since we were not looking for new candidate genes upregulated in APP mice or in our HT22 cell culture system, we did not narrow our analysis only to genes delivered by a general-purpose differential gene expression approach such as DESeq but tested all B2-SRGs. However, based on the reviewer’s comments below, we realize that the paper would benefit by presenting in the main figures only those B2 RNA regulated SRGs that overlap with differentially expressed genes identified by DEseq in each experimental system. This will help to avoid confusion and any misunderstanding that all B2 RNA regulated genes are equally affected in our system, which is not the case and would be an overstatement. We are now presenting in new Figure 2 (2E, 2F) only those B2-SRGs that overlap with upregulated genes identified by DESeq in 6m old APP mice (listed in new Supplementary table 5 in Supplementary file 1) and in new Figure 7 (7D, F) we are now presenting only those B2-SRGs that overlap with upregulated genes identified by DESeq in HT22 cells treated with amyloid beta (listed in new Supplementary table 11 in Supplementary file 1). The conclusions drawn by the new figures remain the same as with the old ones and we believe that this new way of presentation of this data will prevent confusion and potential over-statements. We thank the reviewer for bringing this to our attention. Based also on this reviewer’s minor point 3, we recommend that the old figures that included all B2-SRGs (and not only the differentially expressed ones identified by DESeq) are moved to the Supplement as new Figure 2—figure supplement 3 and Figure 8—figure supplement 2, respectively, so that readers can still get a view of all the data and the transcription dynamics of all B2-SRGs, while we provide both in text and the supplement an explanation about the value as well as limitations of these figures.

Major comments:1) In Figure 1, the authors indicate a strong connection between B2 RNA regulated SRGs and learning and memory. In Figure 2, they identify the SRGs in the hippocampus, please provide a direct comparison of learning and memory associated SRGs and the SRGs they identify in Figure 2 that are significantly upregulated in APP mice in 6 months.

In the revised version of the manuscript we now provide:

i) As a new figure panel (lower panel in new Figure 1E), the number of B2 RNA regulated SRGs that are associated with learning based on our Peleg et al., 2010 paper and as a new Supplementary table 3 in Supplementary file 1, the exact list of these genes.

ii) As a new Supplementary table 4 in Supplementary file 1, the list of all genes that are significantly upregulated in APP mice (6 months).

iii) As a new Supplementary table 5 in Supplementary file 1, the list of those genes upregulated in amyloid pathology (APP 6 months) that are B2-SRGs (expression levels of these genes are presented in new Figure 2E,F).

Per reviewer’s question, we now provide as a new Supplementary table 6 in Supplementary file 1, the list of B2 RNA regulated SRGs that are both learning associated genes and upregulated in 6 month old APP mice. In the text (first two sections of the Results), we provide direct comparisons of the number of genes in each category and their overlap.

2) To better understand the data in the context of hippocampal function, please include functional annotation of SRGs they identified in Figure 2F as they do it in Figure 1 (desirably for each time point, at least for 6M). How many of the SRGs they identify in Figure 1 are part of Figure 2F? Please include functional annotation of significantly upregulated B2 regulated SRGs in Figure 2 and compare them with that of Figure 1.

The number of B2 RNA regulated SRGs in Figure 1 that are part of Figure 2 (in particular Figures 2E,F) is now presented in the new Supplementary table 5 in Supplementary file 1 and also in the text. We now provide as a new Supplementary table 7 in Supplementary file 1 the functional annotation of these genes (see also general comment for this reviewer) and discuss the findings in the text.

We recommend to include only the 6M old mice as this is the time point in which B2 RNA processing was found to differ between WT and APP mice. However, if the reviewer thinks that this is necessary we will add also differential expression lists of other ages as additional supplementary tables.

3) In Figure 3, the authors report that the B2 processing rates are high at the 6M time point at in hippocampi of the APP mice. Please include the levels of unprocessed and processed B2 RNAs in these samples along with this figure, without which it is difficult to gauge the significance of its correlation with SRGs in Figure 2.

We now provide as new figure panels 3E and 3F the levels of processed B2 RNA fragments and unprocessed (full length) B2 RNAs in these samples, respectively, along with the processing ratio which is now labeled as subfigure 3G.

4) What is the % of B2 regulated SRGs that are hsf1 bound in Figure 4C? What is there dynamics in the wild type and APP hippocampi?

Old Figure 4C is now Figure 4A. The exact number of B2 RNA regulated SRGs that are close to Hsf1 binding sites is now presented as a new figure (Figure 4C) and discussed in the text. A list of these genes is provided as new Supplementary table 8 in Supplementary file 1. For genes that are upregulated in APP mice compared to wild type, the difference in Hsf1 binding dynamics between B2 RNA regulated and not regulated genes is now presented as Figure 4—figure supplement 1.

5) What is the distribution of Hsf1 binding sites on (a) non-B2 regulated SRGs and (b) non-SRG genes in hippocampi?

This point is related with point 4. We now present a new panel (Figure 4B) for non B2 RNA regulated genes (listed in Supplementary table 13 in Supplementary file 1) along with the distribution we have in the initial manuscript for all B2 RNA regulated SRGs (now presented as Figure 4A). The direct comparison of these genes is presented in the new Figure 4—figure supplement 1 together with a similar comparison only for genes upregulated in APP mice.

6) In Figure 4D, the 3months old Wt HSF1 levels are high, yet B2 processing (Figure 3E) is low. Please comment.

The reviewer’s comment made us realize that we should include a plot that describes the correlation between Hsf1 levels and B2 RNA processing ration across all sequenced samples. This should reveal whether differences such as those observed by the reviewer affect our conclusion regarding the relationship between these two parameters. We now provide this in the new Figure 8—figure supplement 1, where we found a strong positive correlation between Hsf1 levels and B2 RNA processing ratio. We thank the reviewer for this comment which helped us to substantiate further this relationship.

7) While the authors show in vitro cleavage of B2 RNA by Hsf1, the experiment lacks controls to be conclusive. At least, please include a similar size protein as HSF1 with no-known RNA binding activity and a similar size protein with RNA binding activity as controls in 5A. Please justify the use of PNK as the control protein. Please include the use domain-based deletions of Hsf1 to map the region of HSF1 that is binding and potentially cleaving the B2 RNA. Please include an RNA of similar size and Antisense-B2 RNA to show the specificity of the Hsf1 based cleavage of B2 RNA. Without these controls, the conclusions in Figure 5 cannot be substantiated.

The endogenous ribozyme activity of B2 RNA compared to other control RNAs has already been shown in two previous works but we will also include the relative controls here by providing control incubations with other RNAs. We will also include the incubations with additional control proteins as suggested by the reviewer. We are currently performing these experiments and will include them in the revised version. PNK is used as a control protein because it is an RNA binding protein that is used in the construction of our short RNA libraries and we wanted show that short RNA seq data are free of such confounding factors that could potentially generate artificial fragments. We now include this information in the text.

We feel that the application of domain based deletions for Hsf1, while it would add additional information on the exact biochemistry underlying B2 RNA processing though Hsf1, is beyond the scope of this manuscript. In the current manuscript we are just focusing on the fact that Hsf1 can accelerate B2 RNA processing in vitro and not on the mechanism how this happens. This should be addressed in our opinion on a separate manuscript.

8) The authors should show that the incubated APP peptides are taken up by the cells (experiments in Figure 5F and Figure 6).

These figures are now labelled as Figure 6C and Figure 7, respectively.

That’s a very interesting point and we thank the reviewer for this comment. Multiple studies have shown that toxicity after incubation by amyloid beta is mediated mainly by cell surface receptors, which through cell signalling leads to the response to cellular toxicity that induces stress genes such as Hsf1.

Nevertheless, APP peptides may enter the cell, and the reviewer’s questions raised the possibility that oligomers entering the cell could have a direct impact on the stability of the B2 RNA. In that case, providing evidence that the amyloid enters the cell would be important if we had indications that amyloid beta interacts directly with B2 RNA. We did test this and we found no direct effect of amyloid beta on B2 RNA, so the processing in our case is not induced by oligomers that may have entered the cell. We were planning to present this information in a different manuscript, but if the reviewer or editor thinks that it would be beneficial for the paper, we could present this as supplement figure that shows that amyloid beta incubations with B2 RNA do not induce further processing beyond what Hsf1 causes. For the moment we just present this in Author response image 1:

9) Please provide the list, functional annotation, and % of the SRGs upregulated upon incubation with APP in HT22 cells in comparison to 6month old APP mice. Comment on learning-related Genes.In the revised version, we now provide and mention in the text the following data:

i) a list of genes upregulated in HT22 cells during amyloid toxicity upon incubation with amyloid beta (new Supplement table 9 in Supplementary file 1),

ii) a list of genes according to point (i) that are common with genes upregulated in APP mice (new Supplementary table 10 in Supplementary file 1),

iii) the list and number of B2-SRGs that are upregulated in HT22 cells during amyloid toxicity (the reviewer’s question) (new Supplementary table 10 in Supplementary file 1). We mention in the text the gene numbers and also the genes that are common in all three lists.

iv) Functional annotation of genes of point (iii) (new Supplementary table 12 in Supplementary file 1),

We also mention in the text the limitations of our comparisons between the in vivo model of amyloid pathology (APP mice) and the in vitro cell culture model of amyloid toxicity (HT 22 cells) and we clarify that the cell culture model is used just as a simulation of the effect of amyloid beta in gene pathways associated with response to cellular stress and the role of Hsf1 on B2 RNA processing.

10) The authors should show the efficient downregulation of Hsf1 (protein) upon anti-Hsf1 LNA transfection.

In the revised version, in addition to the RNA-seq data we provide a second confirmation at the mRNA level with an independent method (RT-qPCR) in new figures 4E and 7B (lower panel). We are currently performing the protein extractions and will provide a WB or an Elisa in the revised version.

11) Please present the total B2 RNA levels for conditions in Figure 6C.

We now provide as new supplementary figure (Figure 8—figure supplement 1B and C) the levels of processed B2 RNA fragments and the total levels of unprocessed full length B2 RNAs of these samples that relate to old Figure 6C (now labeled as Figure 7C)

12) Hsf1 levels are not significantly downregulated in Control cells which were inoculated with the reverse APP peptide. Please comment.

We assume that the reviewer here refers to the lack of reduction in Hsf1 levels in the cells inoculated with the reverse peptide and the anti-Hsf1 LNA. Indeed, this lack of reduction is confirmed also by the new qPCR we performed (new Figure 7B, lower panel, R-ctrl vs R-anti-Hsf1). This should likely be attributed to compensation during non-stress conditions. In contrast, under stress conditions, Hsf1 is heavily used in stress response, which could explain the differences we see as cellular needs surpass the available Hsf1 transcripts due to degradation by the LNA. This is also supported by the new RT-qPCR experiments we have performed for B2-SRGs (new Figure 7E). In agreement with what is known for stress response genes such as immediately early genes (for example FosB), levels of these genes are minimal in both Rctrl and R-anti-Hsf1 conditions and only become activated during stress response. We now discuss this in the text of the revised manuscript.

13) Please compare and contrast the % of genes, the overlap, and the functional distinctions in 6F to that of 5G and Figure 1. What are the genes that are common between Figure 1, and that are specifically upregulated upon Anti-Hsf1 LNA transfection along with 1-42 APP. What is % of the occurrence of B2 binding sites in those genes? What are their functional annotations and what is their connection to learning, memory, and cell survival?

Old Figure 6F is now Figure 7F, while old Figure 5G is now Figure 6C.

This point is discussed in the response to points 1 and 9 of this reviewer. In summary, genes upregulated in our amyloid toxicity model included 25 B2-SRGs (new Supplementary table 11). When testing for enriched terms in these 25 genes, biological processes related with apoptosis, such as regulation of apoptotic process and programmed cell death were at the top of the list (new Supplementary table 12) and included, among others, genes such as FosB and Mitf that have been connected with Alzheimer’s disease. Out of the 25 genes that are up-regulated in both mice and our cell culture system, six are B2-SRGs (4932438A13Rik, Fosb, Pag1, Ptprs, Sema5a, and Sgms1) and include a well-known immediate early gene (Fosb), genes associated with sensitivity to amyloid toxicity (Pag1, Sema5a, Sgms1, Fosb), as well as genes associated with p53 (Ptprs, Fosb). All these genes get upregulated in amyloid toxicity (42-Ctrl vs R-Ctrl) but are not upregulated when Hsf1 LNA is applied (42-anti-Hsf1 vs R-anti-Hsf1, no significant difference). This information is now included in the text.

Minor comments1) Please include TPM/ FPKM values for hippocampal markers as control in Figure 2 to do justice to the hippocampus specific RNA seq conducted by the Authors.

To our understanding, the reviewer here suggests the testing of well-known hippocampal markers in our mouse data as controls to confirm that they are indeed hippocampus specific. We have selected as reference markers, the genes employed by the Allen Brain Atlas RNA-sequencing project and we provide a comparison of their data in hippocampal cells with our data from mouse hippocampus. This is now presented as new Figure 2—figure supplement 1.

2) In Figure 2D the authors show that B2 RNA regulated SRGs in the 3 months' wild type mice are significantly high. P53 has been reported to be high in young wild types hippocampus, but not SRGs in my opinion. The authors should comment on this.

Old Figure 2D is now Figure 2E. We now mention the reviewer’s comment particularly in the Discussion and cite a landmark review article in Neuron journal by Michael Greenberg regarding the role of stress response genes, such as FosB, early during development.

As to prevent any confusion, we have also replaced SRGs with B2-SRGs since we tested only B2-SRGS in our study.

3) In Figure 2F, under the 6m APP condition, the replicate 3 looks substantially different from the other replicate. This can significantly impact the analysis and conclusions made. Either remove that replicate and present the analysis without it or please provide a valid explanation. To make the data more valid, please provide hierarchical clustering of the entire data, the non-B2 regulated genes and the B2 regulated SRGs.

We now provide in the new Figure 6—figure supplement 1C a PCA plot, which includes 6m APP mice vs. their WT counterparts and HT22 cells, and shows that this variability is within the biological replicate variability we can expect in these models. To substantiate this further, we have constructed the correlation matrix of the RNA-seq data of both WT and APP 6 month old mice in the new Figure 6—figure supplement 1D. As shown in this matrix, all APP mice clearly correlate with each other and not with their WT counterparts.

In the initial manuscript the heatmaps of former Figure 2 were indeed provided with hierarchical clustering of the entire data and also included non-B2 RNA regulated genes. This data is included now as Figure 2—figure supplement 1.

In Figure 2C RNA seq data is represented in TPM while its FPKM in Figure 2D.

Figure 2D is now Figure 2E, while Figure 2C remains labelled with the same number.

Given that TPM already includes scaling of the data, it is unsuitable for the averaging of the gene expression levels of multiple genes (B2-SRGs) used in the boxplots of Figure 2. This does not apply in the case of single genes as in Figure 2C (p53) or in the heatmap where each gene is presented in a separate row. This explanation is now included in the Materials and methods section.

Figure 2: the number of replicates in the case of 3-month-old wild types only 2. Please specifically denote it and comment why only 2 replicates are provided.

During the hippocampal RNA extractions, the RNA of one of the three 3m old mice had very low RIN scores, which could be a confounding factor for the short-RNA-seq. As this happened some months after the hippocampal extractions, we did not have any other 3 month mice of the same cohort used for the behavioral and IHC studies. Thus, we decided to include only two replicates in this condition. Since the results presented in the current study focus mainly on 6 month old mice, we expect the impact to be minimal. We include this note in the Materials and methods section.

4) Considering that p53 and SRGs are significantly upregulated in 6months in the APP model, it would be great if (allowing that these samples are still available) the authors can include a staining for apoptotic markers, for example, Active Casp3 or similar. This will allow us to better gauge the gene expression changes presented by the authors especially regarding SRGs.

Unfortunately, we do not have these slides but in the revised version we will provide qPCR data for some of these markers.

5) Under subheading: Hsf1 accelerates B2 RNA processing, 3rd paragraph when the authors comment on known hsf1 binding sites on SRG genes, please correct from: Increased Hsf1-binding was found…. "To the increased number of hsf1 binding sites were found", unless the authors would like to show increased Hsf1 binding by performing ChIP-seq for Hsf1 in the hippocampus at least at the 6-month time point between Wt and APP mice.

We have changed the text accordingly.

Reviewer #1 (Significance (Required)):B2 RNAs, encoded from SINE B2 elements has been directly implicated in stress response by its inherent ability to bind RNA Pol II and suppress stress response genes (SRG) in homeostatic conditions. However, upon stimuli, B2 RNAs are cleaved and degraded, resulting in the release of RNA pol II and upregulation of SRGs. Previous work from the senior author identified PRC2 component EZH2 to be the B2 RNA processing factor, cleaving B2, and releasing POL2. SRGs are upregulated upon stress, for example in age-associated neuropathologies like Alzheimer's disease (AD). Considering that the hippocampus is a primary target of amyloid pathologies as well as since SRGs are suggested to be key for the function of a healthy hippocampus, the authors set to understand the role of B2 RNAs that are linked to SRG regulation in the mouse hippocampus with amyloid pathology. They use disease-relevant in vivo and in vitro models combined with unbiased RNA seq data analysis for this endeavor, which indicates the potential relevance of B2 RNAs in APP mediated neuronal pathologies in mice as well as identifies Hsf1 as the factor cleaving B2 RNAs in the hippocampus.The work is interesting and identification of Hsf1 as the processing factor for B2 RNAs in the hippocampus is significant. I would like to credit the authors for their elegant in vivo experimental design in Figure 2.Reviewer 2:Reviewer #2 (Evidence, reproducibility and clarity Required):Summary:This manuscript follows from previous work by the corresponding author showing that SINE-encoded B2 RNAs function as regulators of the expression of stress response genes (SRGs). Specifically, stimulus triggers the processing of repressive B2 RNAs that are bound at the SRGs, thereby activating SRG transcription. In this work, the authors investigate whether a similar mechanism might be controlling the expression of genes in models of amyloid beta neuropathology (i.e. mouse hippocampi from an amyloid precursor protein knock-in mouse model, and a cell culture model of amyloid beta toxicity). They performed RNA-seq in these models. Their data show a correlation between the progression of amyloid pathology, expression of genes thought to be regulated by B2 RNA, and the processing of B2 RNA. In addition, they show biochemical data supporting a role for Hsf1 in enhancing the processing of B2 RNA. Knockdown of Hsf1 also reduced B2 RNA processing and the expression of SRGs.Major comments:1) In the RNA-seq data one cannot distinguish between Pol III transcribed B2 RNA and Pol II transcribed B2 RNA (typically embedded within introns and UTRs of mRNAs). The models they present, and the structures they show, clearly imply regulation by Pol III transcribed B2 RNA. However, there is no way to know that the short B2 RNAs they sequence aren't coming from degraded mRNAs. This needs to addressed. Minimally, in writing as a caveat of their model. Ideally, it would be addressed experimentally.

That’s a very interesting point, as it implies that the regulatory role of B2 RNAs may extend from PolIII transcribed B2 RNAs into B2 RNAs embedded into mRNAs (likely nascent ones) that may be also under the same endogenous ribozyme activity of this sequence, suppress PolII and are processed in response to stimuli. The RNA RIN values of our samples were pretty high except one 3m old mouse sample which was for this reason excluded from further analysis. Moreover, during the library construction shorter and longer RNAs have been separated. Thus, any generation of B2 RNA fragment that may have originated from mRNA should be biologically but not technically related and must have happened in the cell before our RNA extraction. To address this point, we now provide a new supplementary figure (Figure 3—figure supplement 2), where we have separated the B2 elements against which we map the RNA fragments into two categories, those that fall within exonic/genic regions and those outside of these regions. Although B2 RNAs are produced by multiple copies in the genome, each copy does harbor multiple SNPs, insertions and deletions, which means that each B2 RNA fragment is mapped to a specific set of B2 elements and not to all of them. In other words, despite multiple mapping a level of spatial specificity is maintained. If the B2 RNAs we map were coming exclusively from either only Pol III B2 elements or mRNA embedded B2 elements, we would expect at least some difference in the distribution of fragments between B2 elements of these two categories, as the second one overlaps with mRNAs. As shown in the new figure 3—figure supplement 2, the fact that distribution models are very similar between the two categories indeed supports the hypothesis that both types of B2 elements may contribute to B2 RNA processing. Most importantly, the profile of B2 RNAs in genic regions shows that B2 RNA processing is not random but follows the same processing rules as B2 RNAs from Pol III promoters. Given the limitations posed by the repetitive nature of B2 RNAs, it remains difficult though to provide an exact number regarding the portion of B2 RNA fragments produced by each category and this is clearly noted in our revised discussion part. However, even the indication that B2 RNAs embedded in mRNAs may also play an important role in our model provides a new perspective that should be investigated further in future studies.

2) The direct regulation of SRGs by B2 RNA was not shown in their model systems for amyloid beta neuropathology. Rather, the authors' used the genes identified in their prior studies as B2 RNA-regulated, which I believe were in the NIH3T3 cell line. Given that transcription is highly cell-type specific, these genes might not be regulated by B2 RNA in mouse hippocampi or their cell culture model, despite the correlations shown. This needs to be addressed. Ideally, a targeted approach to show that transcription of even a couple genes in their system is indeed regulated by B2 RNA would provide stronger support for their conclusions.

We agree with the reviewer and we now provide a new figure (Figure 6D-F) with the targeted approach that this reviewer proposed. In particular, we have tested whether fragmentation of full length B2 RNAs is in connection with activation of target genes also in our biological system (HT22 cells) as it did in NIH/3T3 cells in our Cell paper. We now show in new Figure 6 that this is indeed the case.

3) The following bioinformatics analyses would strengthen their conclusions. This should be straightforward to do because it involves data they already have, and perhaps analyses they have already have performed.a) Regarding the plot in Figure 3A (lower panel). The same plot should be shown for the 3m old and the 12m old APP mice (i.e. not just the 6m data). This would show the specificity of processing B2 RNA and that it indeed correlates with disease progression.

We now provide this plot as new supplementary figure (Figure 3—figure supplement 1). It shows that increased B2 RNA processing coincides only with the active neurodegeneration phase at 6 months and not the terminal stage.

b) Regarding the plots of B2 RNA processing rate. This value could increase either due to more short RNAs or less full length RNA. Which is it for the 3m, 6m, and 12m APP mice? Showing the short and long B2 RNAs as boxplots (as opposed to only the processing rate) would address this and also provide additional insight into the regulation involved. The same applies to the data in Figure 6. (As an aside… do the authors mean processing ratio as opposed to rate? I'm not clear where the time component is coming into play to call this a rate.)

Old Figure 6 is now Figure 7.

We now provide all these figures that show that increase in processing ratio at 6 months is mainly due to increase in the processed fragments and not a decrease in full length B2 RNAs. For APP mice these are new Figures 3E and F, and for HT22 cells , these are new Figure 8—figure supplement 1B and C.

c) The random genes in Figures 2E and 6E are plotted as heat maps, but statistical significance is hard to see. What do boxplots of the random genes look like, and is the significant difference between 6m old APP and 6m old WT then lost?

Old Figure 2E is now new Figure 2—figure supplement 3C, while old Figure 6E is now new Figure 8—figure supplement 2C. We now provide these boxplots in new Figure 2—figure supplement 3B and Figure 8—figure supplement 2B.

4) It is interesting that B2 RNA self-processing is enhanced by both Ezh2 and also Hsf1. It would strengthen the data to perform a control with a protein prepared more similarly to the Hsf1 (rather than PNK) to confirm that the enhanced B2 RNA breakdown is indeed attributable to Hsf1 and not a contaminant in the protein prep. Similarly, the authors should provide information on which RNA was added as the negative control for Hsf1-stimulated breakdown (i.e. the ~80 nt RNA).

This point is also discussed in reviewer 1 point 7. The ribozyme endogenous activity of B2 RNA has been shown already in two previous studies that performed incubations with control RNAs and proteins. We are currently preparing and will provide these additional incubations as anew supplementary figure in the revised manuscript.

Minor comments:1) Regarding the GO analyses in Figure 1 (panels B, C, and D). I wasn't clear whether the authors are showing all statistically enriched terms, or only those relevant to neuronal processes and learning. I recommend showing a supplemental table with all terms that have an adjusted p value below a specified cut-off (e.g. 0.05).

The statistical threshold used was an EASE score of 0.05 and all presented terms were above this threshold. In the initial manuscript we filtered only the top 5 terms in tissue enrichment and the top 10 terms for GO Biol process and Cell Compartment that had passed the threshold. We now provide all the terms that passed the threshold as a new Supplementary table 2 in Supplementary file 1, including gene counts, exact gene numbers and related statistics.

2) The authors show several figures that are not new data (2B, 4A, 4B, Supplementary figures 1 and 2). I think it would be more clear if these data were summarized and referenced in the Results, rather than shown.

Old Supplementary figures 1 and 2 that were results of previous studies or web resources directly available (such as Human Protein Atlas) have been now removed and they are now just referenced in the text. Old Figures 4A and 4B have been removed from the main figures but may be helpful to the readers if they are still available in the Supplement (currently as Figure 4—figure supplement 1A and B), as not all users are familiar with the RNA-seq browsing tools of Allen Brain Atlas resources.

Regarding Figure 2B that contains data from our previous study on this exact cohort of mice: If the reviewer and the editor agree we recommend that it remains in the main figure (with the appropriate image credit citations), as it provides in an efficient way the clear connection between amyloid load and our results at the molecular level, and, most importantly, it clearly draws a line in amyloid pathology progression between 3m old and 6m old, that agrees with our findings in the RNA-seq data of these mice.

3) In Figure 3A the schematic shows that B2 is 155 nt, the plots in Figures 3A,B,C show B2 RNA is 120 nt, and Figure 5 shows the RNA is 188 nt. Can the authors please clarify these differences?

The full length of B2 consensus sequence is 188nt and this is the one we use for the in vitro experiments. However, the structure of the B2 RNA has been resolved only for the first 155nt by the Kugel lab, and this is the only publicly available structure that we can reference in our figures.

For the mapping of 5’ends of short fragments in Figure 3A we have used the same range tested in our Cell paper to maintain consistency of the results. The reason why this 120nt threshold was selected in the Cell paper was to exclude artifacts from short RNAs mapping partially in our metagene as well as downstream of those B2 elements that are shorter from the consensus sequence. We now explain in Materials and methods section these differences.

4) In the Materials and methods section, the sequence of the g block template didn't contain the T7 promoter sequence that was used as the forward primer for PCR amplification?

We have now included this sequence in lower case.

5) In Figure 6B, why were Hsf1 levels not decreased in the R treated cells after treatment with the LNA?

Old Figure 6B is now new Figure 7B.

Please see response to reviewer 1, major point 12.

Reviewer #2 (Significance (Required)):The models presented for the regulation of stress response genes (SRGs) in amyloid beta neuropathologies are compelling. As are the correlations they found between the progression of amyloid pathology, expression of genes thought to be regulated by B2 RNA, and the processing of B2 RNA. This is a unique direction of research for brain disease and represents an interesting conceptual advance. Most prior studies in this area use common model cell lines, and this lab seems well-positioned to unravel the proposed molecular mechanisms in neuronal systems.

We appreciate the encouraging comments made by this reviewer.

Reviewer 3:Reviewer #3 (Evidence, reproducibility and clarity (Required)):This manuscript describes a regulatory mechanism involving Hsf1 and B2 RNAs in the control of stress response genes (SRGs) during amyloid induced toxicity. In particular Hsf1, upregulated in 6m old APP mice and in HT22 cells treated with beta amyloid peptides, is shown to stimulate the B2 RNA destabilization leading to SRGs activation. While in healthy cells this upregulation can be reverted once the stimulus is removed, the pathological condition fuels the circuitry leading to p53 upregulation and neuronal cell death. The authors previously described the same mechanism acting during cellular heath shock response but in this case the protein identified as trigger of B2 RNA destabilization and SRGs activation was EZH2 (Zovoilis et al., 2016).Indeed, the first part of the manuscript describes additional analyses of the previous data that prompts further investigation on the potential role of B2 RNA in AD condition. Nevertheless, it is not clear how the prior findings obtained in not biologically related cellular models might be used to obtain helpful indication of B2 RNA neuronal activity.

We thank the reviewer for this comment. Indeed, the current study’s main aim was to expand the findings of our previous work on the role of B2 RNA in cellular response to thermal stress in NIH/3T3 cells to other types of cellular response to stress, in our case to amyloid toxicity and the resulting amyloid pathology in neural cells. Response to thermal stress (Heat Shock) has been used for years as a basic study model for cellular response to stress. Proteins and gene pathways initially identified in heat shock have been subsequently shown to play identical pro-survival roles in other biological systems and there are studies showing the role of Hsf1, heat shock related proteins and cell stress response pathways in neural cells and the mammalian brain (we will provide these references in the revised version). For example, pathways such as the MAPK pathway and early response genes, that constitute the basis of response to heat shock, have been shown in studies by us and others to be activated and play a critical role in hippocampal function. Thus, examining the role of B2 RNA in the context of neural response to stress constituted a natural continuation of our previous study in NIH/3T3 cells. The fact that the list of B2 RNA regulated SRGs was found to be highly enriched in neuronal tissue terms and cellular compartments related to neuronal functions plainly confirms the close relationship among cellular response pathways in the two biological systems. Due to these facts we were compelled to investigate in more detail our previous findings also in a neural cell model. However, as discussed in point 2 of reviewer 2, the initial manuscript did not confirm the direct control of B2 RNA on expression of target genes also in our cellular model. This information is now part of the new Figure 6 and we thank both reviewers for bringing this to our attention.

The research fields of non-coding RNAs and neurodegeneration are attractive and challenging and, in my opinion, the molecular circuitry involving B2 RNAs might add important insights for understanding beta amyloid toxicity and neuronal death; however, the data provided are not in the shape making the manuscript suitable for publication: some controls are missing, the way the experiments are presented is not easy to follow and more importantly the authors does not provide any data (tables or lists) of the NGS experiments and the study lacks validation of them. Therefore, in my opinion the manuscript needs a profound revision before to be considered for publication in Review Commons.

Based on this reviewer’s and the other reviewers’ suggestions we now provide additional controls, detailed tables and gene lists, and qPCR validation of these results. We have also substantially revised the text in the first section of the Results and beginning of the Discussion, to make our rational for testing B2SRGs more clear and easier to follow.

Major concerns:1) The first paragraph of the Results is entirely dedicated to reanalyze the data previously published by the same group (Zovoilis et al., 2016). However, this is not adequately explained. In line with this, the Table 1 is not required since the data are already provided by Zovoilis et al., 2016, unless the authors handled the data using additional new criteria that have to be explained.

We now explain our rational for using this data in more detail in the text. Please see also response to the general comment of this reviewer and response to the next point.

In the Zovoilis et al., 2016, study, the data presented did not include the list of regulated genes in a direct way but as part of the annotation of the B2 CHART peaks. This may pose difficulty to non-experts to extract the gene list from that data and we thought to include them as separate gene list here so that readers can directly use it for their analysis. Nevertheless, if the reviewer or the editor think that the list is redundant, we can surely omit it.

Moreover, Zovoilis and colleagues (2016) focused on SRGs regulated upon heat shock and using NIH/3T3 and HeLa cell lines, therefore, it is difficult to me understand how, searching for "cellular function connected with B2 RNA regulated SRGs", the list resulted enriched of neuronal tissue terms or cellular compartments related to neuronal functions. Please clarify this point since the following analyses are based on these findings.

Neural pathologies, such as amyloid pathology in brain, are often connected with cellular stress due to proteotoxicity. The ability of neural cells to respond to proteotoxicity challenges is connected with various molecular mechanisms, including stress related proteins that were firstly described in the context of heat shock. Thus, both contexts (heat shock and amyloid toxicity) refer to cellular response to stress, which explains why genes identified to be regulated during stress response in NIH/3T3 cells constitute part of the basic stress response toolbox that neural cells have also been described to possess. We have now modified the text accordingly to make our rational more clear.

2) In Figure 1F there is no arrow indicating that some of the SRGs regulate directly miR-34 as stated in the main text. Moreover, it is more appropriate to replace SRGs with learning-associated genes both in the figure and in text (second paragraph of the Results) since Zovoilis and colleagues focused on them. Finally, they did not show in their manuscript the rescue of p53 expression mediated by mir-34; indeed, for miR-34-p53 regulatory axis Zovoilis and colleagues referred to Peleg et al., 2010 and Yamakuchi and Lowenstein, 2009. Please fix all these concerns.

We have restructured the figure as suggested by the reviewer and made clear the distinction between learning genes and B2 RNA regulated SRGs (B2-SRGs) from the two different studies. In connection with point 1 of reviewer 1, we believe that new Figure 1E, that includes the exact number of B2-SRGs that are learning associated, will represent more efficiently and accurately the data. We have also corrected in the text the citation regarding miR-34c and p53 in both the Introduction and first section of the Results (last paragraph).

-The Figure 1A and Figure 1F are wrongly indicated at the end of the sentence "….levels of these genes are normally downregulated in 6m and 12m old mice compared to 3m old mice (p=0.02 and p=0.04, respectively)"; please correct this point.

The error has been corrected.

3) Regarding Figure 2:a) Since three mice for each condition have been used for the RNA seq analyses, please provide a blot with the Principal Component Analysis (PCA).

Please see also response to minor point 3 of reviewer 1. We provide the PCA plots for WT and APP mice in the new Figure 6—figure supplement 1 and we also provide a comparison of the six month old mice with the HT cell samples as well as a correlation matrix for 6 month old mice in the same figure.

b) Figure 2F comes first of Figure 2E in the text, however, I suggest to move this latter to supplementary material.

Old Figure 2E has now been moved to supplementary material as new Figure 2—figure supplement 1C and we also provide in a boxplot the exact gene expression levels as new Figure 2—figure supplement 1B.

c) In general, this study lacks validation of the RNA-seq results. Western blot and/or qRTR-PCR to verify the variation of p53 and of some selected SRGs have to be provided.

In the current revised version we already provide qPCRs for p53 and Hsf1 in APP mice and we will include additional genes in the final version.

d) It is also not clear how the authors defined SRGs in the hippocampus: do they correspond to learning associated genes described by in Zovoilis et al., 2011 or to B2 RNA H/S regulated genes by Zovoilis et al., 2016?

The way we presented B2 RNA SRGs in the results with regard to learning associated genes was indeed unclear. We now present the distinction between the two gene categories and their relationship as a new Figure 1E panel and we also provide detailed gene lists of common genes and the exact numbers (please see also response to Review 1, major point 1).

APP 12 month old mice show the sever phenotype of the terminal AD-like pathology, however this does not correlate with significant SRGs and B2 processing increase. Can the author make a comment on this?

That’s a very important point and we thank the reviewer for raising this point. We now comment on this in the Discussion part explaining how our findings are characteristic of the initial active neurodegeneration phase of amyloid pathology rather than more terminal stages.

4) Regarding Figure 5:a) A gel with no-protein control for the time course of panel B was cited in the text but missing among the panels. Moreover, the time course shown in the graph in 5C does not correspond to the one in 5B.

Indeed, the no-protein control time line should refer only to panel C and not to B, we have now corrected the text. Nevertheless, we now present in the new Supplementary Figure 5 the gels, based on which the graph in panel C was calculated, including also the gel with no protein timeline.

The time course shown in the initial 5C had been mislabeled. It has now been corrected. We apologize for this and we thank the reviewer for bringing this to our attention.

b) 5G indicates that four samples for each condition have been analysed by RNA-seq, since they do not seem to be homogeneous please provide a PCA analysis together with the validation by qRT-PCR of a selected group of deregulated genes.

Old Figure 5G is new Figure 6C. PCA analysis for these samples is now provided in Figure 6—figure supplement 1 and qPCR validation of a number of these genes is provided in new Figure 7E.

Moreover, it is not clear whether all the genes shown in the heatmap or a number of them, as stated in the text, were found upregulated in 6m old APP mice. Please clarify this point and modify the figure and the text accordingly. A Venn diagram showing the overlap between genes upregulated in 42vsR treatment and those upregulated in 6m old APP mice might help the comprehension of the experiment.

Please see response to reviewer 1, point 9. We now provide as new supplementary tables the exact overlapping lists and mention these numbers in the text.

5) Regarding Figure 6 (now labeled as Figure 7):a) The evaluation of the levels of Hsf1 mRNA and protein upon LNA transfection is missing for both R and 42 treated HT22 cells. From TPM in panel B, Hsf1 downregulation seems to have been more effective in 42 than in R condition. This would mess up the interpretation of the data.

We now provide qPCR data for Hsf1 gene expression levels which confirm the ones from the RNAseq. The reason why Hsf1 downregulation seems not to affect the R condition is discussed in our response to reviewer 1, major point 12, and the respective explanation is provided in the revised text.

b) Again, in this case any validation of the RNA seq data is provided (any B2 regulated SRGs).

Now, we provide qPCR data for these genes in Figure 7B and new Figure 7E

c) Panels E and F should be swapped or panel E moved to supplementary material.

Panel E is now moved to supplementary material as new Figure 8—figure supplement 2C.

6) In a previous paper the authors discovered B2 RNAs as a class of transcripts bound to EZH2 and this interaction leads to B2 RNA destabilization in heath shock (H/S) condition. The authors also conclude that the genes controlled by B2 RNAs may not overlap with the ones controlled by Hsf1 during H/S. The author should make a comment on this explaining why during H/S B2 RNAs work independently from Hsf1 and on different target SRGs while, during beta amyloid stress ,the two act together on the same SRGs. Moreover, as shown for EZH2, Hsf1-RIP experiment should be performed in order to confirm the direct involvement of Hsf1 in the SRGs-B2 destabilization.

In the last two paragraphs of our Discussion we indicate that B2 RNA regulation is a new process implicated in the response to stress in amyloid pathology but certainly not the only one. We have revised the text in this part accordingly in the revised version to prevent any confusion. We are currently performing a series of RIP-seq experiments with various antibodies. As, to our knowledge, there is no prior published study performing RIP-seq or CLIP-seq for any tissue using Hsf1 antibodies, the success of this experiment is not guaranteed and depends on the existence of appropriate antibodies.

7) There is any table listing the results of the RNA seq experiments performed in this paper: control vs APP 3-6-12 m old mice and in R vs 42 treated HT22 cells in presence or absence of LNA against Hsf1. Please provide these data.

We now provide these lists as new supplementary tables. Please see response to major points 1 and 9 of reviewer 1.

8) In the Discussion the authors claim that healthy cells are able to restore the expression of Hsf1, SRGs and B2 RNA upon removal of the stress. Since there are evidence for the rescue of SRGs and B2 RNA expression post H/S, no data are available for Hsf1, SRGs and B2 RNA upon the removal of 1-42 beta amyloid peptide. This might be a nice information to add to the manuscript.

This would indeed substantiate further our results in our HT22 cell model. We have now performed this experiment, in which HT-22 cells were removed from the amyloid 42 (and the respective R peptide control) and left to recover for 12 hours before estimating through RT-qPCR the Hsf1 levels ( see graph in Author response image 2, REC corresponds to recovered HT-22 cells). Hsf1 levels in 42-REC have returned to the same levels as in R, p< 0.05 for the difference between 42 and 42-REC, y axis=rel expression levels (A.U.).

**Author response image 2. respfig2:** 

We currently perform the RT-qPCRs of these samples also for B2-SRGs and will include them in the final version as a supplementary figure.

Minor criticisms:– In the Introduction the reference Yamakuchi and Lowenstein 2009, should be added in the sentence: "In contrast, hippocampi of mouse models of amyloid pathology and post- mortem brains of human patients of AD … and neural death (Zovoilis et al., 2011)."

We have now changed the text at that point accordingly and also updated the legend of Figure 1F that also refers to this same study.

– Authors refer to Hernandez et al., 2020 to state that B2 self-cleavage is stimulated by some proteins however, Hernandez and colleagues studied only the effect of EZH2 protein. Please rephrase the sentence accordingly.

Text has been modified accordingly.

– Indicate a reference for the sentence: "……Ezh2, was reported as being responsible for the B2 RNA accelerated destabilization and processing during response to stress."

The respective citation was added.

– The format of many references is not consistent and has to be revised.

We have switched to the Vancouver style. Some references in the legend and Materials and methods sections are referred independently from EndNote in case these text sections have to be moved to supplement in the final version in order to not create inconsistencies with endnote.

Reviewer #3 (Significance (Required)):The research fields of non-coding RNAs and neurodegeneration are attractive and challenging and, in my opinion, the molecular circuitry involving B2 RNAs might add important insights for understanding beta amyloid toxicity and neuronal death. However, this manuscript does not really add technical advances since the authors employed experimental approaches and bioinformatic analyses previously published by Zovoilis and colleagues in 2011 and 2016.

Our aim in the current manuscript was not to introduce a new method or experimental approach but rather to study the mechanisms behind B2 RNA regulation of gene expression in neural cells and particularly in amyloid pathology. Nevertheless, the current study constitutes the first reported short-RNA seq in this tissue and offers for the first time the ability to study B2 RNA processing in this tissue which is not possible with standard small and long RNA-seq.

The reported findings might of interest of an audience of experts in non coding RNAs and neurodegeneration. The area of my expertise almost regards the biology of non coding RNAs from biogenesis to function manly focusing on neuronal and muscular systems both in physiological and pathological conditions.

[Editors' note: further revisions were suggested prior to acceptance, as described below.]

After thorough discussion, the reviewers agreed that the manuscript by Zovoilis et al. is much improved from the original submission to Review Commons and that the authors have addressed many of the original concerns raised by reviewers. However, two major concerns remain, and reviewers agreed to encourage a resubmission addressing these.B2 RNAs encoded from SINE B2 elements have been directly implicated in stress response by its inherent ability to bind RNA Pol II and suppress stress response genes (SRG) in homeostatic conditions. However, upon stimuli, B2 RNAs are cleaved and degraded, resulting in the release of RNA polymerase II (RNAPII) and upregulation of SRGs. Previous work from the senior author of this manuscript identified the Polycomb repressive complex 2 (PRC2) component EZH2 to be the B2 RNA processing factor, cleaving B2 and releasing RNAPII. SRGs are upregulated upon stress, for example in age associated neuropathologies like Alzheimer's disease (AD). Considering that hippocampus is a primary target of amyloid pathologies and given that SRGs are suggested to be key for the function of a healthy hippocampus, the authors set out to understand the role of B2 RNAs that are linked to SRG regulation in the mouse hippocampus with amyloid pathology. They use disease relevant in vivo and in vitro models combined with unbiased RNA seq data analysis for this endeavor, which indicate the potential relevance of B2 RNAs in APP-mediated neuronal pathologies in mice while also identifying Hsf1 as the factor cleaving B2 RNAs in the hippocampus. The work is deemed interesting and identification of Hsf1 as the processing factor for B2 RNAs in the hippocampus is significant.1) Reviewers remain concerned that the bulk of the analyses relies on genes that were identified as B2-regulated SRGs in a prior experimental system (heat-shocked NIH3T3 cells) that is completely different from the amyloid pathology models used here. The authors sought to address this issue in the revised manuscript, but questions remain. Indeed, the new Supplementary tables 4 and 5 in Supplementary file 1 show that of the ~1600 B2-SRGs identified in NIH3T3 cells, only 72 show expression patterns consistent with the regulatory model proposed; moreover, this was using FDR<0.2. How many genes would be left with FDR<0.05? The authors did include important new data in the HT22 cell model showing that mRNA levels for four genes increase after treatment with a B2-targeted LNA. These data are compelling, but a few additional controls are needed. Figure 6F needs negative controls showing qRT-PCR of genes that are not thought to be B2-SRGs. In Figure 6E, it is important that the full length B2 RNA is being detected (i.e. a PCR product ~180 nt) since their model states that these genes are repressed by full length B2 RNA prior to its degradation. The data in Figure 6 support the model that these four genes are under B2 control, but they don't show the relationship with amyloid pathology. What do the expression patterns of the 4 genes in Figure 6F look like in the mouse RNA-seq data (3m, 6m, 12m, WT vs APP)? Does the amyloid beta peptide treatment of the HT22 cells no longer induce expression of these four genes in the presence of the B2 LNA?

We thank the reviewers for this comment. We now provide more clarity and supporting data to address this comment. In particular:

a) Our model does not propose that all B2-SRGs are deregulated in APP mice or that all deregulated genes in APP mice are B2-SRGs. Instead, it shows that a subset of the B2 SRGs is deregulated in amyloid pathology. This may not have been clear in the model described in our initial manuscript (old Figure 8, now Figure 9), in which we have used the general term “Stress response genes” to describe genes under this regulatory mechanism. In the revised Figure 9 model we have now substituted this term with the more accurate term “Target genes” to prevent any confusion and denote that there may be a variety of B2 RNA-targeted genes in different biological contexts. Moreover, our current data addresses B2 RNA-mediated regulation only in the context of amyloid beta pathology and focuses on differentially expressed B2 SRGs based on a statistical modeling approach (DESeq2, a total of 72 genes). Beyond this set of differentially expressed genes, we searched for additional SRGs that are expressed in the hippocampus, yet, they may be associated with other hippocampal functions that are independent from amyloid beta pathology. The expression of hippocampal B2-SRGs would be expected to correlate with B2 RNA processing. As shown in the new Figure 8—figure supplement 1E and the respective table (new Supplementary table 12 in Supplementary file 1) a detailed breakdown of these B2-SRGs with regard to expression in hippocampal cells and correlation with B2 RNA processing reveals 659 B2-SRGs that are sufficiently expressed in our tested hippocampal samples and, of those, more than 63% shows a weak (11%) or strong (52%) correlation between expression levels and B2 RNA processing ratio. This data suggests that, for a large number of the B2-SRGs previously identified in the context of heat shock, correlation between gene expression and B2 RNA processing ratio holds true also in the context of hippocampal cells, independently of the presence of amyloid beta pathology.

b) By applying DESeq2 to statistically model the expression changes observed between 6m old WT and APP mice we obtained 72 B2-SRGs upregulated in 6m old APP mice (listed in Supplementary table 5 in Supplementary file 1). All of those B2-SRGs have a p-value < 0.05, with 34 of them belonging to the group with an FDR value threshold below 5% and the rest 38 to the group with an FDR value at least below 20%. We used a more flexible FDR threshold since expression dynamics of the 72 differentially expressed genes (shown in Figure 2) show an almost identical pattern with those ones of several other B2-SRGs that may have been missed by the DESeq2 (shown in Figure 2—figure supplement 3). This strongly suggests that use of DESEq2 is rather conservative in our case for reasons listed in the respective note in the legend of Figure 2—figure supplement 3. In order to mitigate any concerns for an increased number of false positives in our data due to selection of a higher FDR threshold, in the new Figure 2—figure supplement 2, we now provide validation through RT-qPCR of the DESeq2 results. We have selected 12 of the differentially expressed genes (Figure 2—figure supplement 2A) and tested them through RT-qPCR (Figure 2—figure supplement 2B). The majority of the genes tested have been intentionally selected from the group with the higher FDR threshold (8 out of 12 have an FDR more than 5% and less than 20%). The resulting RT-qPCR data presented in Figure 2—figure supplement 2B are in alignment with the RNA-seq results for the same genes presented in Figure 2—figure supplement 2A. Thus, through the use of an orthogonal validation technique, the differentially expressed genes identified by DESeq2 in 6m APP mice indeed correspond to true positive findings. This data, together with data in Figure 2—figure supplement 3, support further that application of a standard RNA-seq differential expression statistical model such as DEseq2 in our case may indeed be a rather conservative approach. Nevertheless, given the widespread use of DESeq2, we still suggest the use of this approach in the main body of the study in order to facilitate comparison of our data with those from other groups in the future.

c) In order to substantiate further the connection between B2-SRGs previously identified in heat shocked NIH/3T3 cells and B2 RNA destabilization we have increased the number of B2-SRGs tested through RT-qPCR in the B2 KD assay in hippocampal cells (initially 4 genes, now 11 genes). In the new Figure 7C, we now show that, as previously observed in NIH/3T3 cells, also in hippocampal cells, destabilizing B2 RNA leads to increase in the expression of these genes. In addition to these 11 genes, we now include 5 negative controls (non-B2 SRGs) that showed no change during treatment. Among the negative controls we have also intentionally included non-B2-SRGs such as Adcyl1 and Kalrn, both of which are upregulated in amyloid beta pathology, to support further the specificity of our findings (i.e. being upregulated in amyloid beta pathology does not mean regulation by B2 RNA unless being a B2-SRG).

d) Indeed, in Figure 7A we test the unprocessed B2 RNA encompassing the full Pol II binding region (80-132) including all cutting points. The reverse primer is located downstream at position 142. Making a reverse primer beyond the position 142 is not possible as it overlaps with B2 RNAs poly-A repeats. The primers are now listed in the Materials and methods.

e) Data for the expression patterns of all genes in the new Figure 7C (old Figure 6F) that are differentially expressed in mouse APP and WT mice are now presented in Figure 2—figure supplement 2A for all three ages and conditions.

f) “*Does the amyloid beta peptide treatment of the HT22 cells no longer induce expression of these four genes in the presence of the B2 LNA?”.* To our understanding, we are asked whether treating the HT22 cells with amyloid beta and B2 LNA induces expression of these genes. Based on the data presented in the manuscript the amyloid beta treatment induces expression of these four genes (see new Figure 6D) as does the B2 LNA independently (Figure 7). Thus, we are a little unsure on the scope of such as experiment as we feel that inoculating B2 LNA together with amyloid beta would not add any additional information regarding our model. Such an experiment would have been informative if B2 LNA treatment was inhibiting SRG activation as in case of Hsf1 LNA (Figure 8), however, since both the amyloid and B2 LNA activate SRGs, we did not see any reason for combining them in one experiment as we did in case of Hsf1 in Figure 8, in which we combined LNA treatment with amyloid beta treatment.

2) Reviewers remain concerned about the strength of the data regarding the role of Hsf1 in B2 RNA processing (although it seems like the authors are currently working on important control experiments, which might alleviate reviewer's concerns.) The negative controls with recombinant proteins prepared similarly to Hsf1, and with similarly sized control RNAs are critical. In addition, the full gels for these experiments need to be shown so the formation of short B2 RNA products can be evaluated in conjunction with the loss of the full length B2 RNA. This will help distinguish between specific processing controlled by the B2 ribozyme/Hsf1 activity versus non-specific breakdown. Moreover, the sizes of the in vitro processed products should correlate with the 5' end peaks from Figure 3A if the cellular and in vitro processing indeed arises from the same mechanism.

We are grateful to the reviewers for this comment. Although some of these controls had already been presented in our previous studies, including control proteins not inducing B2 RNA processing and control RNAs not destabilized as rapidly as B2 RNA, we fully understand that it is due to the proposed biochemistry of B2 RNA as a potential ribozyme, and the difficulty to distinguish from non-specific RNA degradation, that additional data are required. To this end we respond below to the reviewer’s questions, and additionally we further test the effect of (i) changes in B2 RNA sequence itself and, (ii) denaturation of the Hsf1 protein on B2 RNA processing. In detail:

a) In new Figure 5F-I we now provide the following:

*–* Incubation of a mutant B2 RNA in the presence and absence of Hsf1 and comparison with the same incubations with the original B2 RNA, that showed that acceleration of B2 RNA processing by Hsf1 is sequence-specific and, thus, not the result of non-specific RNA degradation.

*–* Incubation of two control RNAs in the presence and absence of Hsf1 and comparison with same incubations with the B2 RNA, that also confirmed that Hsf1-induced processing is limited to B2 RNA.

*–* Incubation of B2 RNA in the presence of a denatured Hsf1 and a control protein of similar size, that failed to accelerate B2 RNA processing as the native Hsf1 did.

b) In order to exclude non-specific RNA destabilization due to degradation, hydrolysis or B2 RNA endogenous self-cleavage, we have included in Figure 5 a number of control RNA incubations without protein in the same buffer (TAP) and for the same time as the samples treated with Hsf1 or control proteins. Full gels are also included now as two source files.

c) We have now performed short RNA-seq of the in vitro processed B2 RNA and compared the resulted fragments with those observed in vivo, now presented in the new Figure 5—figure supplement 1 A. This data reveals a similar processing pattern between the two conditions.